# Modeling of three-dimensional innervated epidermal like-layer in a microfluidic chip-based coculture system

Jinchul Ahn[1,2,8], Kyungeun Ohk[3,4,8], Jihee Won[1,2], Dong-Hee Choi [1,2], Yong Hun Jung[1,2], Ji Hun Yang[2], Yesl Jun [5,6], Jin-A Kim[1] ✉, Seok Chung [1,5,7] ✉ & Sang-Hoon Lee[4,5]

Reconstruction of skin equivalents with physiologically relevant cellular and matrix architecture is indispensable for basic research and industrial applications. As skin-nerve crosstalk is increasingly recognized as a major element of skin physiological pathology, the development of reliable in vitro models to evaluate the selective communication between epidermal keratinocytes and sensory neurons is being demanded. In this study, we present a three-dimensional innervated epidermal keratinocyte layer as a sensory neuron-epidermal keratinocyte co-culture model on a microfluidic chip using the slope-based air-liquid interfacing culture and spatial compartmentalization. Our co-culture model recapitulates a more organized basal-suprabasal stratification, enhanced barrier function, and physiologically relevant anatomical innervation and demonstrated the feasibility of in situ imaging and functional analysis in a cell-type-specific manner, thereby improving the structural and functional limitations of previous coculture models. This system has the potential as an improved surrogate model and platform for biomedical and pharmaceutical research.

The skin contains a complex network of sensory nerve fibers as a highly sensitive organ; mechanoreceptors, thermoreceptors, and nociceptors. A variety of neuronal subtypes whose cell bodies reside in the dorsal root ganglia (DRG) are densely and distinctly innervated into cutaneous layers[1-4]. These neurons exhibit distinct anatomical localization according to their morphological, neurochemical, and sensory functions. Specifically, large diameter and thickly myelinated Aβ-fibers which detect mechanical stimuli are innervated in the dermis, whereas unmyelinated C-fibers and thinly myelinated Aδ-fibers detect thermal and nociceptive stimuli are innervated both in the epidermis and dermis (Fig. 1a)[3,5-8]. Free nerve endings of peptidergic or non-peptidergic C-fibers are mainly located close to keratinocytes in the spinous layer or granular layer of the epidermis, providing the structural basis for functional interaction such as synaptic-like contacts[9-12]. Consistently with this physical contact, recent studies have shown that sensory nerve fibers in the skin can express and release nerve mediators such as neuropeptides that signal the skin, including calcitonin gene-related peptide (CGRP), substance P (SP), and neurokinin A[13,14]. In addition, it has been shown that skin cells themselves, such as keratinocytes, can also release neurotrophic factors that determine nerve fiber density, morphology, axon growth, and neuropeptide levels[14-16]. These reports have extended the biological significance of nerves to

[1]School of Mechanical Engineering, Korea University, Seoul 02841, South Korea. [2]Next&Bio Inc., Seoul 02841, South Korea. [3]R&D center, Humedix, Co., Ltd., Seongnam 13201, South Korea. [4]Department of Bio-convergence Engineering, Korea University, Seoul 02841, South Korea. [5]KU-KIST Graduate School of Converging Science and Technology, Korea University, Seoul 02841, South Korea. [6]Drug Discovery Platform Research Center, Therapeutics and Biotechnology Division, Korea Research Institute of Chemical Technology, Daejeon 34114, South Korea. [7]Center for Brain Technology, Brain Science Institute, Korea Institute of Science and Technology (KIST), Seoul 02792, South Korea. [8]These authors contributed equally: Jinchul Ahn, Kyungeun Ohk. ✉e-mail: lamon0516@korea.ac.kr; sidchung@korea.ac.kr

sensation as well as other biological skin functions, suggesting their physical and pathological correlations with several skin diseases. They naturally led to the development of several in vitro models to understand skin-nerve interactions [15–20].

However, the traditional 2D coculture systems have failed to spatially locate a cell or cell portion (e.g., the axon and cell body of a neuron) and to selectively analyze and probe specific cells. Cultured keratinocytes also suffer from morphological and functional limitations [15,17,18,21]. The keratinocytes in vivo have existed in proliferating states at the basal layer of the epidermis, and they undergo differentiation to form a spinous, granular, and cornified layer (Fig. 1a)[22]. 3D transwell culture platforms and microfluidic chips have been developed and further technologically improved by designing 3D culture conditions for epidermal morphogenesis and cell-customized compartmentalization for co-culture[13,19,20,23–26]. In the 3D transwell insert culture system, a full-thickness human skin model with histological and functional properties that exhibit physiological similarity to in vivo skin was developed, but a reliable innervated skin model has yet to be reported[20,23–25,27–31]. A recently reported sponge-based co-culture model, like the transwell insert culture, also failed to mimic the anatomical distribution of intra-epidermal free nerve ending and axon patterning, notwithstanding the well-differentiated epidermal layer (Supplementary Table 1)[20]. The advantages of microfluidic chips, commonly referred to as lab-on-a-chip or cell chips[19,32], have made them attractive candidates to replace traditional experiments, by reducing the sample volume and the cost of reagents, and providing investigators with substantially precise control and predictability of the spatiotemporal dynamics of the cell microenvironments and fluids[19,32]. In particular, the advantages of the spatiotemporal control allow researchers to closely recapitulate in vivo functions (both normal and disease states) by integrating several well-understood components into a single in vitro chip. However, reliable skin-nerve interactions and communication in the anatomically innervated epidermis have not yet taken advantage of microfluidics because they are based on the structure of vertically stacked systems, such as transwell insert cultures [16,19,33,34].

This work presents a microfluidic model for coculture and analyzes 3D interactions of keratinocytes and sensory neurons (SN) in vitro. Technically, a slope-air liquid interface (slope-ALI) culture was applied to provide an air contact necessary for epidermal differentiation without additional devices, demonstrating advancements in keratinocyte development in terms of epidermal differentiation, cell layering, and barrier function compared to conventional microfluidic chip systems using planar liquid culture. It was also shown that the hydrogel-based multi-channel system recapitulated the cellular/subcellular compartmentalization and cell-cell/cell-matrix interactions, leading to the physiologically relevant organization of the innervated epidermal-like layer and enabling functional analysis in a cell-type-specific manner, such as the in-situ permeability assay of the epidermis and sensory transmission assay initiated by topical stimulation to epidermal keratinocytes. Finally, we modeled epidermal keratinocyte-sensory neuron crosstalk in our platform under hyperglycemic conditions mimicking acute diabetes and demonstrated its feasibility as a model for investigating the underlying mechanisms of the pathological condition.

## Results

### Microfluidic chip for keratinocyte-sensory neuron co-culture
In order to mimic the physiologically innervated epidermal anatomy (Fig. 1a), we designed and fabricated a hydrogel-incorporated microfluidic chip (Fig. 1a, modified from the previous design[32]). The chip contains physically comparted four cell culture and analysis units (channels), including one soma channel for neurons and another epidermal channel for keratinocytes. The soma and epidermal compartments are connected by two 500 μm width axon-guiding microchannels that function as a physical barrier to confine neuronal soma in the soma channel, allowing axons to grow toward the epidermal channel by neuronal sub-compartmentalization. Posts arranged between channels help the spatial distribution of multiple hydrogels and/or cells. Keratinocytes loaded into the epidermal channel grow on one side of extracellular matrix (ECM) hydrogel and interact only with axons but not with neuronal soma, enabling localized axon-keratinocyte interaction studies like in vivo physiology (Fig. 1a and c). This cellular compartmentalization allows two independent cells to be conducted on a single device maintaining cellular identity and function, and also allows to selectively analyze and/or probe specific cells and cell portions (e.g., the axon and cell body in a neuron) that cannot be done in 2D and transwell insert co-culture system (Fig. 1c). Each axon-guiding microchannel is individually filled by physiologically relevant ECM hydrogel, i.e., type 1 collagen, acting as a layer of acellular dermal ECM, yet exclusively without fibroblasts[22,35,36]. After seeding DRG neuron cells (in the soma channel) and human epidermal keratinocytes (HEK, in the epidermal channel) sequentially, the medium in the keratinocyte channels was emptied and the cell-filled chip was tilted to maintain above 30 degrees tilt to mimic the air-liquid interface (slope-ALI culture), a common and critical microenvironment for the skin cell differentiation (Fig. 1b, 1d, and Supplementary Fig. 1 and 4a–c)[31,37,38]. The developed microfluidic chip enables various imaging, biochemical and functional analyses such as axonal response testing and integrity/permeability tests, which can be conducted directly on the innervated epidermis-on-chips, thus improving the limitations of conventional transwell insert culture or microfluidic culture system (Fig. 1c).

### Fine-tuning of axonal patterns in the multi-compartment microfluidic chip
To pattern nerve fibers from the soma channel through hydrogel into the keratinocyte layer, we first optimized the composition and concentration of connected ECM hydrogel components, depending on the context of the cells it comes into contact with, respectively DRG SN and keratinocytes (HEK) (Fig. 1a). Three combinations of hydrogel conditions were examined for SNs culture on the microfluidic chip: type 1 collagen at a concentration of 2 mg/ml (COL2), type 1 collagen at a concentration of 2 mg/ml with 10% laminin (COL2L), and type 1 collagen at a concentration of 1.5 mg/mL with 10% laminin (COL1.5 L) (Fig. 2). Primary DRG SNs from E15 rats were loaded to the soma channel and cultured for 1 week. Whereas neurites of SNs were dispersed irregularly (without a constant axonal pattern) on conventional PDL/laminin-coated 2D plate, axons in our microfluidic chip crossed ECM channels and reached epidermal channels forming axon-only network layers. Soma of the SNs was aggregated in the ECM hydrogel and the axons were 3D aligned with directional elongation through the hydrogel in the opposite direction (axon/epidermal compartment) over the incubation time (Fig. 2a). The width of the 3D neurites in the hydrogels of the microfluidic chips was significantly thicker than that of the 2D plate (Fig. 2f). In 3D, the number of neurites per soma was also higher, forming bundle-like structures (Fig. 2a). Length and outgrowth of neurites were inversely proportional to hydrogel concentration, regardless of the presence of laminin (Figs. 2b, c and 2d, e and Supplementary Fig. 3)[39,40]. Interestingly, laminin was found to remold the width and angle of neurites; their initial angle in the ECM hydrogel was widely aligned under all ECM conditions at DIV 2, but that in the ECM presenting laminin was gradually narrowed and straightened at DIV 6 (not significant but trending, Fig. 2a and g). Taken together, COL1.5 L is an optimal condition for guiding axonal elongation, preserving only soluble factor-mediated communication (minimal cell migration) and resulting in dense axonal network formation.

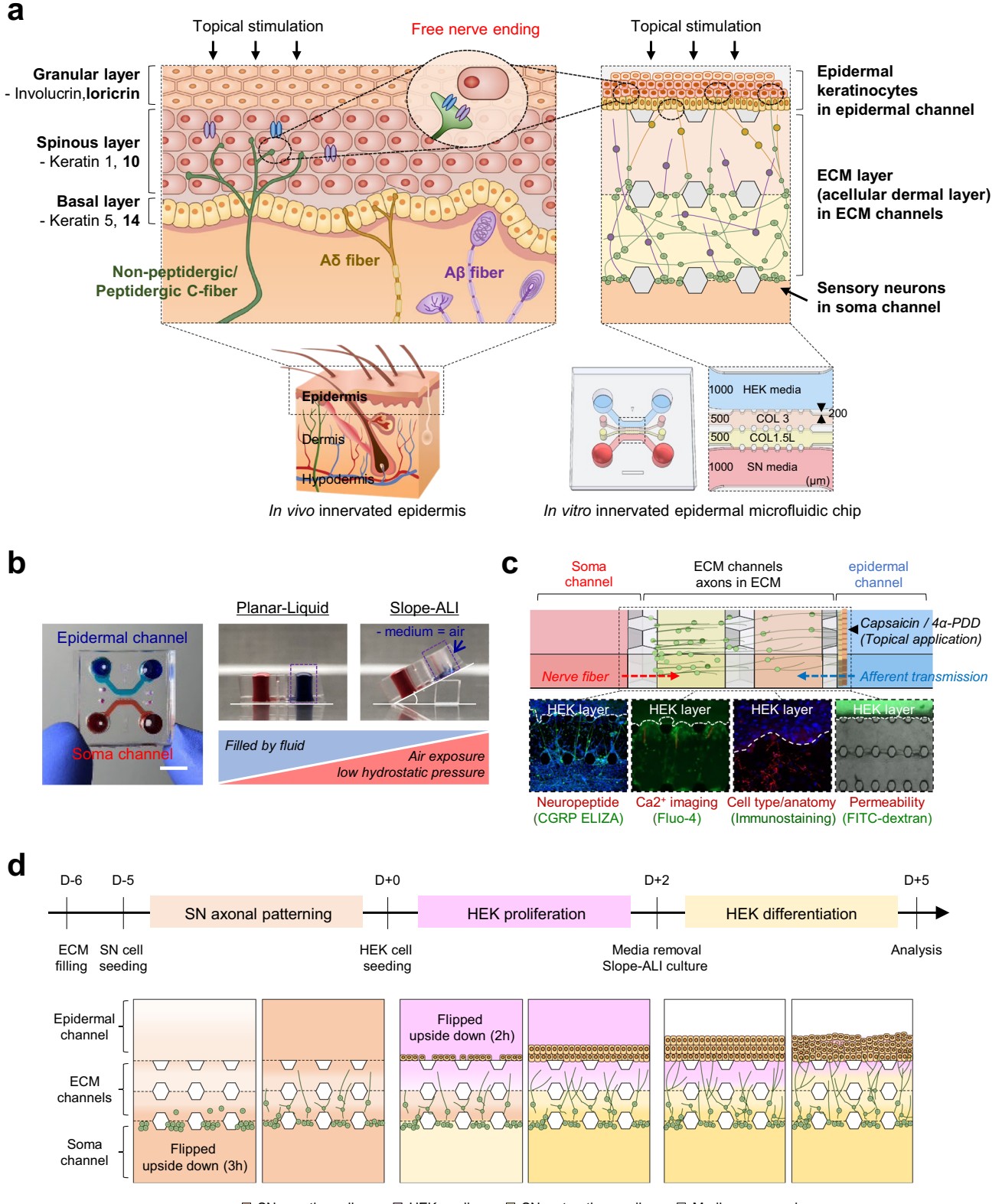

**Fig. 1 | Microfluidic platform and culture system for sensory neurons-keratinocytes co-culture. a** Schematic illustration and design of human skin anatomy (left) and the innervated epidermal chip to coculture sensory neurons and keratinocytes (right). Schematic design of the innervated epidermal chip compartments (right lower). HEK; human keratinocyte, SN; sensory neuron, COL 3; collagen I at 3 mg/ml concentration, COL 1.5 L; collagen I at 1.5 mg/mL with 10% laminin, Scale unit; μm. **b** Top view of the microfluidic chip (left) and experimental concept of slope-based air-liquid interface (ALI) method for epidermal development (right, longitudinal vertical section view). Each cell channel was marked with a different color dye. **c** Cell-type-specific assays for the innervated epidermal chip. **d** Experimental workflow of cell seeding and culture for generating the innervated epidermal chip.

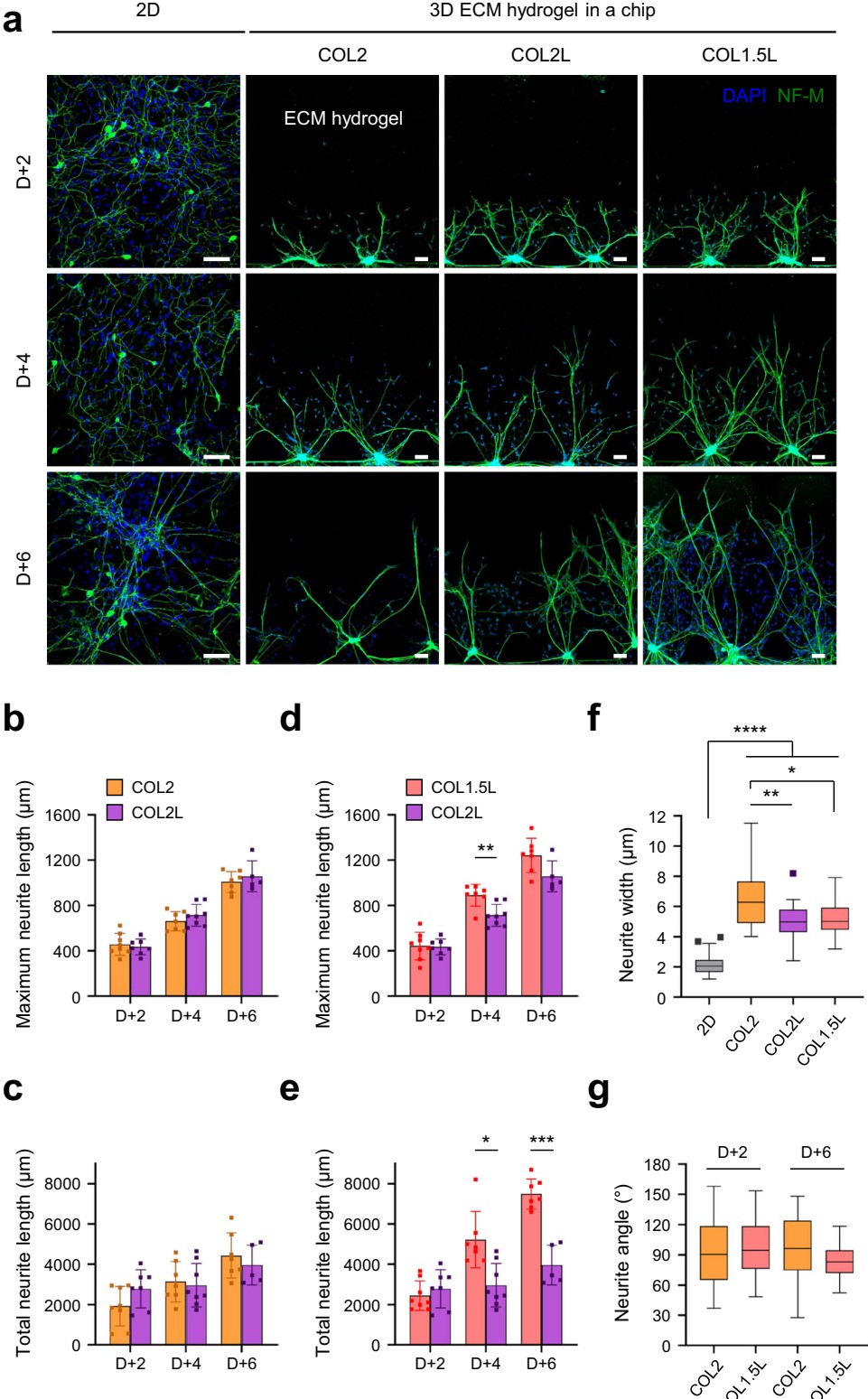

## Epidermal development in air-liquid interfacing micro-fluidic chip

Formation of epidermal layer structure and its barrier function are mainly accomplished by proliferation and differentiation of keratinocytes[22]. Basal keratinocytes adjoin the underlying ECM which forming the dermal-epidermal junction (DEJ). The ECM provides a specific niche that mediates mechanical and chemical signals to keratinocytes through cell-ECM interactions[22,41]. In the developed

microfluidic chip, the dermal ECM layer was formed by two neighboring (double-layered) ECM hydrogels, one type 1 collagen at a concentration of 3 mg/mL to support seeded keratinocytes to form the stable epidermal-like layer, and the other COL1.5 L to support neuron's easier adhesion and migration, similarity to the DEJ microstructure (Fig. 1a). We monitored morphologies of the mono-cultured keratinocyte layer under HEK medium in the epidermal channel and varied the medium in the soma channel (HEK medium, SN medium, and their

**Fig. 2 | Optimization of 3D extracellular matrix (ECM) hydrogels for axon patterning of sensory neurons in a microfluidic chip. a** Representative fluorescence images of elongated nerve fibers of sensory neurons in microchannels for each ECM condition. NF-M; neurofilament M, green, DAPI; nuclei, blue. COL 2; collagen I at 2 mg/ml concentration, COL 2 L; collagen I at 2 mg/mL with 10% laminin, COL 1.5 L; collagen I at 1.5 mg/mL with 10% laminin. 2D; conventional monolayer culture method. Scale bars; 100 μm. **b–g** Quantitative analysis of axonal changes according to ECM conditions of the chip. Maximum (**b**, **d**) and total neurite length (**c**, **e**) of sensory neurons at each time point after culture ($n = 5$–8 ROIs, at least 10 neurites were measured in each ROI, COL1.5 L(d4) vs COL2L(d4)

$**p = 0.0014$, COL1.5 L(d6) vs COL2L(d6) $p = 0.1211$ for maximum neurite length, COL1.5 L(d4) vs COL2L(d4) $*p = 0.0126$, COL1.5 L(d6) vs COL2L(d6) $***p = 0.0006$ for total neurite length, 2 independent replicates). Box plot of the neurite width (**f**) of a sensory neuron 6 days after culture ($n = 19$ ROIs, 2D vs COL2, COL2L, COL1.5 L $****p < 0.0001$, COL2 vs COL2L $**p = 0.0041$, COL2 vs COL1.5 L $*p = 0.0119$, 2 independent replicates). Box plot of neurite angles (**g**) of sensory neurons 2 days and 6 days after culture ($n = 36$–40 ROIs, 2 independent replicates). One-way ANOVA, Bonferroni's multiple comparisons test. Data are mean ± SD, $*p < 0.05$, $**p < 0.01$, $***p < 0.001$, $****p < 0.0001$. Box plot shows median and 75th and 25th percentiles, and whiskers show minimum and maximum values.

1:1 mixture). Interestingly, the SN medium (other than the medium in the epidermal channel) applied to the soma channel made the keratinocyte layer stable and thick (Supplementary Fig. 4).

To model the sensory innervation to the epidermis[11], we first adapted the slope-ALI method to induce epidermal differentiation (Fig. 1b)[31,37,38,42]. Our slope-ALI method rapidly initiates ERK activation and the proliferation of keratinocytes than the planar-liquid method, resulting in thicker epidermal-like layers (Figs. 3a–c, 3k, l, and Supplementary Fig. 6). This method developed multicellular epidermal differentiation such as the basal (cytokeratin 14+, K 14), suprabasal (cytokeratin 10+, K 10), and granular (loricrin+, for late-stage differentiation) cells compared to the planar-liquid method: which consists mainly of K14+ keratinocytes but few K10+ and loricrin+ cells (Fig. 3d–f). The K14+ and K10+ keratinocytes of the slope-ALI method formed the suprabasal layer just above the basal layer like human epidermal tissue, showing a structurally more organized cell layer than the planar-liquid method (Fig. 3g). Under slope-ALI conditions, undulating micropatterned structures were noticed in the keratinocytes layer, like Rete ridge (RR) in natural human skin which has never been noticed in current tissue-engineered or 3D skin equivalents (Supplementary Fig. 7)[41]. The keratinocyte layer in slope-ALI condition was tortuous but tightly interconnected showing a strong barrier function to 3.984 kDa FITC-conjugated dextran, consistent with more intense and continuous distribution results (Fig. 3g). It also showed enhanced blocking for the diffusive transport from the epidermal channel to the soma channel, consequently facilitating cell-type-specific functional analysis (Fig. 1c). Taken together, these results indicate that our slope-ALI culture can accelerate the proliferation of keratinocytes and their aligned layering during differentiation, reconstituting the tortuous layered epidermal keratinocyte layer.

## Histological features of the innervated epidermal-like layer in the microfluidic chip

To recapitulate the physical contact between epidermal keratinocytes and SN, we co-cultured keratinocytes and SN in a microfluidic chip and evaluated the structural and functional characteristics in a cell-type-specific manner as described in Fig. 1c. First, morphological features of co-cultured SNs were characterized by immunostaining. PGP 9.5 + sensory neurites arise from the soma channel, penetrate the double-layered ECM hydrogels, move toward the tortuous epidermis, and finally terminate around the epidermal-keratinocyte interface similar to intraepidermal nerve endings (Fig. 4a–c). Fibers from mono-cultured SNs were smoother and less branched, while the outgrowth of nerve fibers from co-cultured SNs relaxed near the keratinocytes, producing thinly branched ends; divided into multiple strands of the free nerve endings (Fig. 4b and 4d, e). Please note serpentine morphology of co-cultured SN's free nerve endings in epidermal keratinocyte layer correlating interfaces of keratinocytes, ending at variable heights with slightly varicose and branched patterns in the basal and spinous layers of the epidermis (Fig. 4b–e and 4g–i).

In the experiments, we found that NF200+ A-fibers (myelinated A-fibers) were more predominant than CGRP (peptidergic unmyelinated C-fibers) or IB4 (non-peptidergic unmyelinated C-fibers) positive neurons (Fig. 4f and 4h, i). NF200+ A-fibers from co-cultured SNs have

morphologically thinner and longer than those from mono-cultured SNs and usually terminate in dermal ECMs falling short of the epidermal layer (Fig. 4f and 4h, i). CGRP+ peptidergic neurons were significantly more in co-cultured SNs and were mainly confined in the region under the epidermal layer, some terminated within the epidermis as free nerve endings. Whereas IB4+ non-peptidergic fibers from mono-cultured SNs had more quantity in ECM hydrogel. Although IB4+ neurons from co-cultured SNs migrated through ECM hydrogel relatively longer than those from mono-cultured SNs, they did not innervate into the deep epidermis (granular layer) and failed to recapitulate the complete anatomy of the native skin (Fig. 4f and 4h, i)[8–10]. Co-culture of SNs influence the development of epidermal keratinocytes in terms of morphogenesis and differentiation (Fig. 4j, k). When innervated, the epidermal-like layer grew on, not invading into the hydrogel, and presented enhanced alignment of K14, K10 and Loricrin (Supplementary Figs. 7a, 8a and 9a). In addition, the co-cultured epidermal keratinocyte layer showed a slight improvement in barrier function against 376.27 Da FITC-sodium (Fig. 4l, m). The co-culture of keratinocytes and SN in our slope-ALI microfluidic chip was proved to recapitulate cellular and histological structures of the innervated epidermis more successfully than conventional 3D transwell insert culture or microfluidic culture system.

## Functional integration of the innervated epidermal-like layer in the microfluidic chip

Innervation of the epidermal layer in the developed microfluidic model noticed that sensory neuron innervation influenced the epidermal development by increasing epidermal thickness and differentiation (Fig. 4). Structural and functional similarities acquired by the model enabled the study of 3D interactions of skin cell components, including functional cross-talk between keratinocytes and neurons during innervating epidermis. Consistent with the previous reports[13–15,43], cell-cell contacts between keratinocytes and neurons were observed in developing epidermal-like layers (Figs. 4b and 5a). Sensory nerve endings sprouting into the epidermal layer were associated with growth-associated protein 43 (GAP43), indicating that the interaction permissive to the outgrowth of neurons occurred spatiotemporally (Fig. 5a)[44]. Levels of CGRP and SP were found to be increased when SN cultured with keratinocytes (Fig. 5b, c), implying paracrine communication by neuron-derived soluble mediators contributing to epidermal integrity[33]. Sensory neurites seemed to form intimate physical interactions with keratinocytes during innervated epidermal development in our microfluidic model.

The model could present a cue for the crosstalk between keratinocytes and SN by cutaneous nociception in normal (healthy) conditions. Sensory-free nerve endings, nerve fibers of innervated sensory neuron, has been known to be a major cutaneous nociceptor. Epidermal keratinocyte also acts as a primary nociceptive transducer, expressing functional sensory receptors and releasing neuroactive substances which specifically activate nociceptive SN to ultimately elicit pain[12,26,45–47]. To assert the nociceptive transduction (Fig. 1a), we treated sensory receptor-specific agonists in the epidermal channel based on the expression of TRPV1 (transient receptor potential vanilloid 1) and TRPV4 (transient receptor potential vanilloid 4) in the

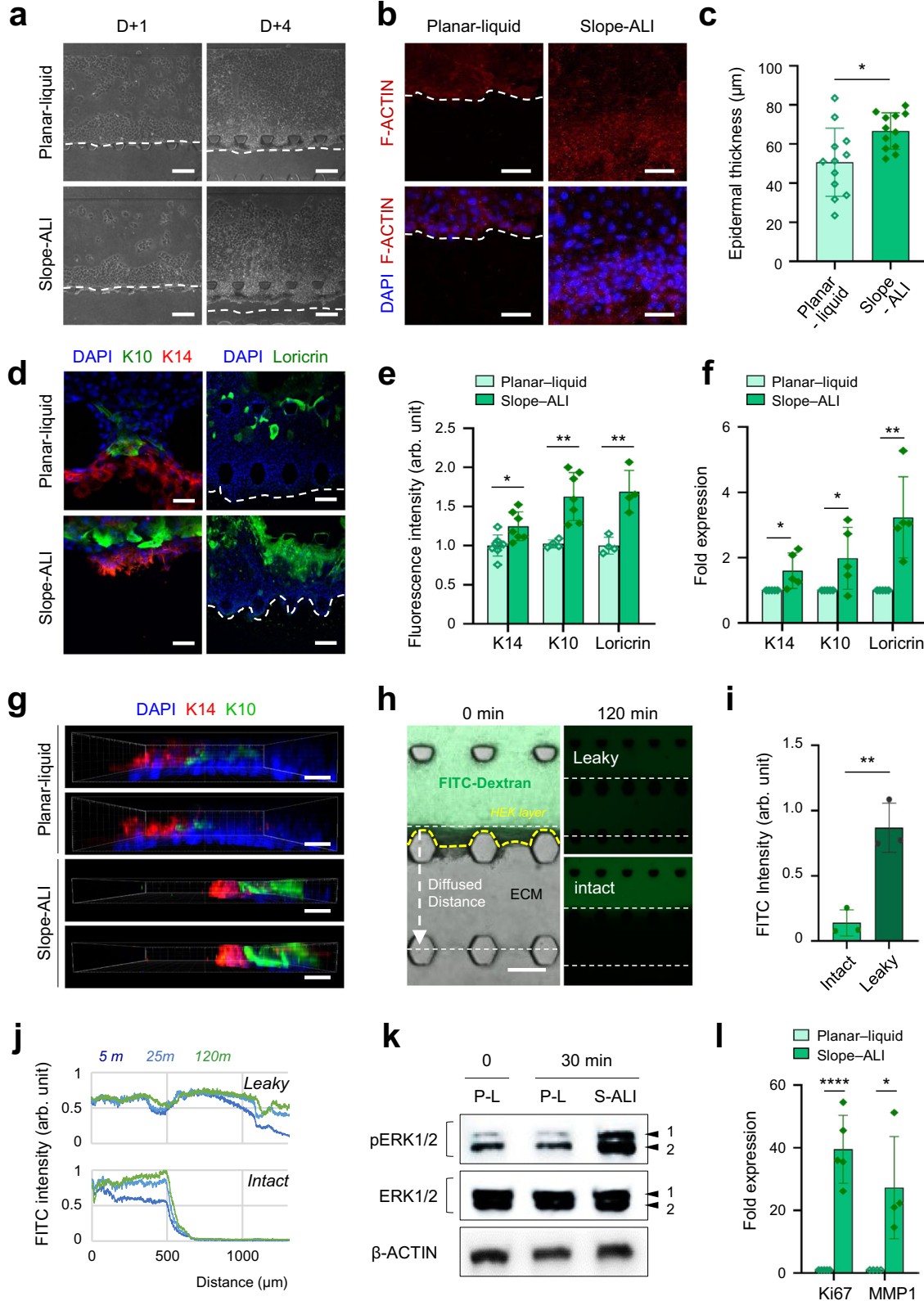

epidermal-like layer (Fig. 5d). Topical applications of capsaicin (for TRPV1) or 4α-PDD (for TRPV4) to epidermal keratinocytes sequentially activated innervated neurons and caused the calcium-dependent release of CGRP from CGRP⁺ neurons (Fig. 5g–j). Single-cell calcium imaging verified that the sensitivity and activity of co-cultured neurons in the epidermal-like layer were enhanced (Fig. 6n and 6p). The SN in the epidermal-like layer seemed to be more sensitive and active than

mono-cultured ones. The enhanced sensitivity of SN by epidermal integration can explain the increased CGRP by topical treatment of capsaicin. The increased number of CGRP⁺ TRPV1⁺ fibers and TRPV1⁺ fibers from co-cultured SN could also be the reason for the increased CGRP. However, the topically applied capsaicin had a low chance of directly initiating the nociceptive response of TRPV1⁺ SNs, due to the strong and intact barrier function of the innervated epidermal-like

**Fig. 3 | Advanced epidermal development on a slope-ALI microfluidic chip.**
**a** Representative bright-field images of the epidermal layer 1 and 4 d after human keratinocytes culture using conventional planar liquid (planar-liquid) or slope-based ALI (slope-ALI) methods on a microfluidic chip (3 independent replicates). Scale bars; 100 μm. **b** Immunofluorescence images of the developed epidermal layers stained with F-ACTIN (red) 5 d after culture on a microfluidic chip. DAPI (blue). Scale bars; 100 μm. **c** Quantification of the epidermal thickness ($n = 12$ ROIs, 3 ROIs per device *$p = 0.0105$, 2 independent replicates). **d** Representative immunofluorescence images for keratin 14 (K14, red), keratin 10 (K10, green), and loricrin (green) in planar-liquid or slope-ALI cultured epidermal layer. DAPI (blue). Scale bars; 50 μm. **e, f** Quantification of fluorescence intensity ($n = 4–7$ devices, planar-liquid vs slope-ALI *$p = 0.0229$ for K14, **$p = 0.0012$ for K10, **$p = 0.0032$ for loricrin, 2 independent replicates) (**e**) and RNA level ($n = 5$ devices, planar-liquid vs slope-ALI *$p = 0.0391$ for K14, *$p = 0.0494$ for K10, **$p = 0.0038$ for loricrin, 2

independent replicates) (**f**) in the epidermal layers cultured with planar-liquid or slope-ALI on a microfluidic chip. **g** 3D confocal images of K14/K10 layer development of the keratinocyte layer (3 independent replicates). Scale bars; 50 μm. **h–j** Permeability of planar-liquid and slope-ALI culture epidermal layers. The distribution images (**h**), time-lapse intensity plot (**j**), and its normalized fluorescent intensity (**i**, at 120 min) of 3.984 kDa FITC–dextran at the interface region of the white dashed line between the ECM hydrogel and epidermal keratinocyte layer in the chip ($n = 3$ devices, **$p = 0.0041$, 2 independent replicates). Scale bars; 200 μm. **k** Immunoblotting of ERK phosphorylation. ERK1/2; anti-total ERK1/2, pERK; anti-phospho ERK1/2. **l** qPCR analysis of ki67 and MMP1 expression in epidermal keratinocytes 24 h after each culture ($n = 5$ devices, ****$p < 0.0001$ for Ki67, *$p = 0.0181$ for MMP1, 2 independent replicates). Data are mean ± SD, *$p < 0.05$, **$p < 0.01$, ***$p < 0.001$, ****$p < 0.0001$. Two-tailed $t$-test.

layer to FITC-sodium comparable to capsaicin (Fig. 4l and 5j, and Supplementary Fig. 8). The developed microfluidic model can demonstrate structural and functional integration of SN in the keratinocyte layer for transmitting afferent information, and protecting capacity for the integrated neurons from the direct impact of topically applied stimuli. The SN also appear to be more active and sensitive when integrated with the keratinocyte layer.

## Mimicking hyperglycemia-induced diabetic neuropathy

Diabetic neuropathy occurs in patients with impaired glucose regulation and is typically characterized by sensory symptoms including pain[48,49]. Though the etiology of diabetic neuropathy is complicated and not fully understood, it is proposed that dysfunctions of intraepidermal nerve fibers under their cutaneous microenvironmental change by hyperglycemia in diabetes may play a significant role[43,48–50]. The developed innervated epidermal-like layer in a microfluidic chip was applied to evaluate the pathophysiological mechanisms of intraepidermal nerve fibers and epidermal keratinocytes in the development and progress of hyperglycemia-induced cutaneous neuropathy by simulating hyperglycemia (Fig. 6a, b).

The effect of a high concentration of glucose (100 mM) on the survival, apoptosis, and oxidative stress of SN was first investigated by staining with a marker for caspase 3 activation or cellular reactive oxygen species (ROS) level. A high glucose environment had no statistically significant effect on neurons' survival and apoptosis regardless of the epidermal-like layer's presence (Fig. 6c). However, it increased cellular ROS accumulation in SN, implying the induction of oxidative stress in the neurons similar to the previous reports (Fig. 6b and d)[49,51]. To determine whether this oxidative stress is structurally and functionally linked to SN, TRPV1+ intraepidermal nerve fibers anatomically distributed in the epidermis were analyzed. The fibers are robust nociceptive SN in the skin having a strong association with clinical disease-associated pain conditions[52–55]. The total number of TRPV1+ neurons had no significant difference under high glucose conditions (Supplementary Fig. 9c), but the length and the number of intraepidermal nerve fibers were significantly decreased. Axonal outgrowth and epidermal innervation of SN seemed to be particularly inhibited (Fig. 6e–g)[54,55]. The high glucose environment did not significantly affect apoptosis, proliferation, and the total thickness of the epidermal-like layer (Fig. 6h, i, and supplementary Fig. 9a, b). However histological analyses revealed that basal and spinous cells were regularly ordered and closely aligned in the epidermal-like layer and became irregular and loose in the high glucose condition (Fig. 6i–l). Keratinocytes in the co-cultured epidermal-like layer showed an altered morphology, being larger and swollen when treated with glucose (Fig. 6i). The expression patterns of K 14 and K 10 showed hyperglycemia perturbed the physiological epidermal differentiation. Added glucose markedly decreased the K10 expression ratio over K14 expression (Fig. 6k, l), and even impair RR morphogenesis and epidermal barrier permeability (Fig. 6i and m).

When capsaicin was topically treated to innervated epidermal-like layer, Ca²⁺ influx responses occurred in epidermal neurons at a single-cell level (Fig. 6n–q). Hyperglycemia reduced intraepidermal SN, but conversely, afferent transmission from epidermal keratinocytes to SN was slightly increased (Fig. 6n). The impaired barrier function of the epidermal-like layer under high glucose conditions increased penetration of topical substance and might induce the direct response of TRPV1+ fibers beneath the epidermal-like layer. The result can simulate the susceptible skin of diabetic patients due to the barrier defects, and provide a possible mechanism for early features of acute hyperglycemia/prediabetes without electrophysiological evidence of nerve damage or sensory dysfunction such as neuropathic pain behavior in diabetic patients despite the loss of intraepidermal nerve fibers. Hyperglycemia is responsible for the aberrant structural development of innervated epidermal-like layers by changing cellular communications, implying the possibility of abnormal functional integration.

## Discussion

Understanding the complex communications and interactions among various cells and neighboring microenvironmental components in the skin is essential for R&D and industrial applications, but challenges have existed in reconstituting 3D structures of cutaneous innervation in vitro and developing analysis tools in a cell-type-specific manner. The in vitro model can be not only an alternative to animals but also an approximation for various human skin diseases and side effects of other diseases on the skin. This paper describes a microfluidic co-culture system to form 3D innervated epidermal-like layers and its qualitative improvements in applicability, reliability, and complexity compared to previous microfluidic co-cultures. Precisely regulated spatial features and co-culture parameters allow compartmental patterning of neurons and epidermal keratinocytes, forming an organized innervated epidermal keratinocyte layer, being clearly visualized in microfluidic format. The microfluidic protocols allow, first, fluidically isolated culture for the two cell populations in distinct patterns or indirect juxtaposition on the same plane of medium and ECM hydrogel (Fig. 1a). Second, the protocols allow temporal and selective analysis and/or probing of specific cells, for example in situ Ca²⁺ response monitoring and epidermal permeability measurement for the topically applied substances. Finally, we hope to mention the benefit of the sloped ALI culture on the epidermal-like layer with low hydrostatic pressure, similar to but slightly different from the conventional insert culture assay. The sloped ALI protocol has the capacity for inducing a mechanical niche for the epidermal-like layer, which is certainly mechanosensitive and may sense compressive stress caused by tilting, and transduce it into physiological biochemical signals such as ERK1/2 cascade and MMPs as previously reported[41,56]. ECM remodeling and internal force made by the proliferation of keratinocytes can help keratinocytes migrate and form bifurcated RR structures, similar to in vivo skin[41]. The RR has physiological significance in strengthening dermal-epidermal

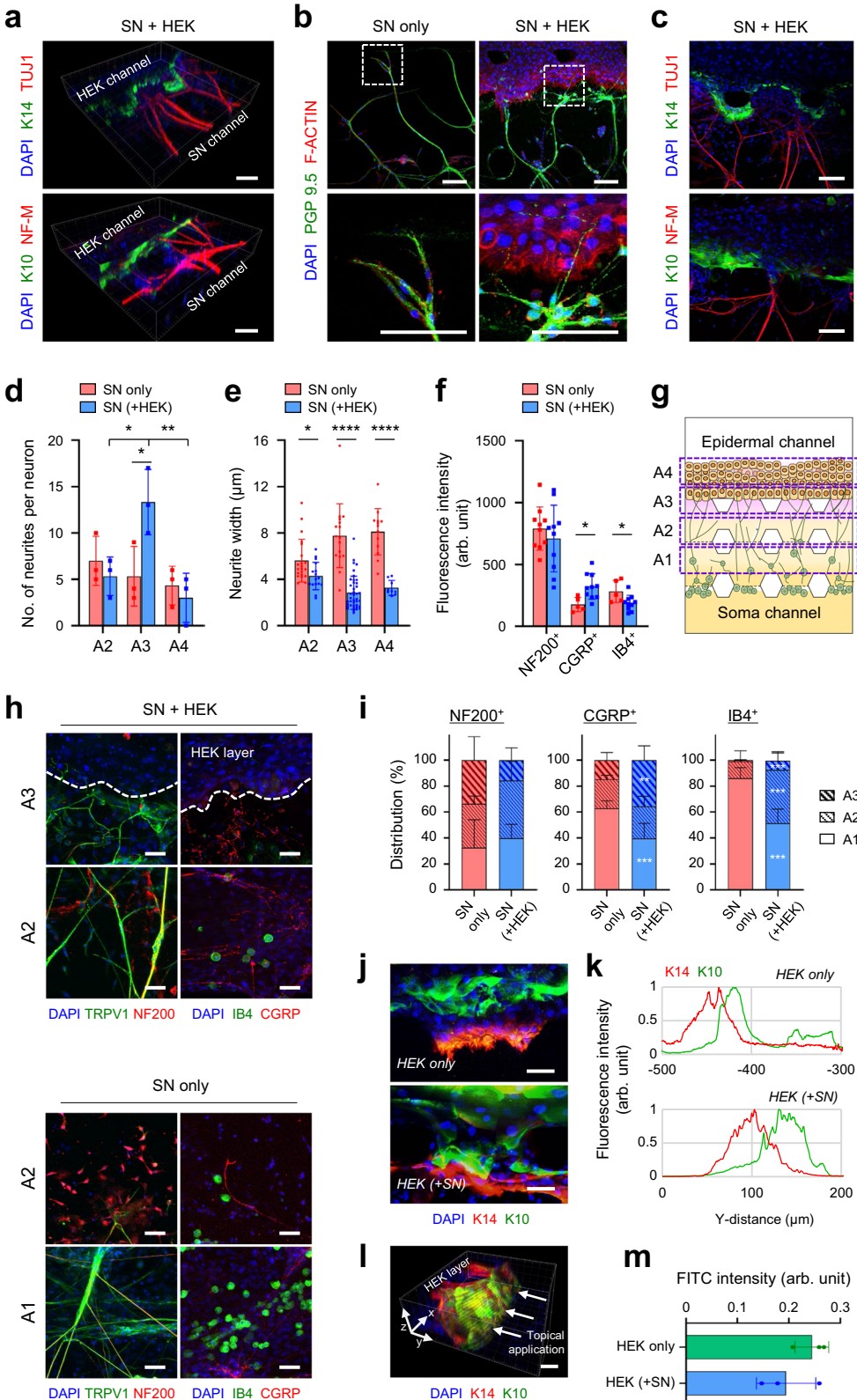

connectivity and improving keratinocyte differentiation by increasing the surface area of the DEJ. Pillars designed in the microfluidic chip to compartmentalize ECM hydrogels might also contribute to the formation of the RR [57].

Another interesting achievement of the developed chip is the spatially distributed various types of SN in the acellular dermal ECM layer and intraepidermal free nerve endings in the epidermal-like layer,

recapitulating the physiological histology of human skin. Nerve and skin cells formed spatiotemporal and physical contacts, representing their functional crosstalk and naive alignments. The chip could successfully model the interactions reciprocally contributed to the development and maturation of innervated epidermis-like layer and afferent transmission of topical stimuli from epidermal keratinocytes to SN. The complex organization of the epidermal-like layer intimately

**Fig. 4 | The structural complexity of the innervated epidermal-like layer in the microfluidic chip. a, c** 3D confocal images of innervated epidermal-like layer for K10, K14 (green) and TUJ1, NF-M (red) (2 independent replicates). Scale bars; 100 μm. **b** Immunofluorescence images for PGP 9.5 (green) and F-ACTIN (red) in SN only or in the SN + HEK group. Magnifications (bottom) of the region highlighted in the white dashed box (top) (2 independent replicates). Scale bars; 100 μm. **d, e** Morphological quantification of sensory neurons along the regions. The number of sensory neurites ($n$ = 3 independent replicates, SN + HEK vs SN only *$p$ = 0.0437 for A3, A2 vs A3 *$p$ = 0.0301 and A3 vs A4 **$p$ = 0.0097 for SN + HEK) (**d**) and the width of sensory neurite bundles ($n$ = 8–39 ROIs, SN + HEK vs SN only *$p$ = 0.0109 for A2, ****$p$ < 0.0001 for A3 and A4, 3 independent replicates) (**e**) in SN only or in the SN + HEK group. **f–i** Comparison of sensory neuron types by quantifying the fluorescence intensity of NF200$^+$, CGRP$^+$, or IB4$^+$ cells between SN only and SN + HEK groups. Quantitative analysis of the total amount ($n$ = 5–10 ROIs, 2 ROIs per device, SN + HEK vs SN only *$p$ = 0.0121 for total CGRP, *$p$ = 0.0323 for total IB4, 2 independent replicates) (**f**) and spatial distribution (**h, i**) of neuron types along the regions (**g**) ($n$ = 5–10 ROIs, CGRP ratio of SN + HEK vs SN only ***$p$ = 0.0004 for A1, **$p$ = 0.0018 for A3, IB4 ratio of SN + HEK vs SN only ****$p$ < 0.0001 for A1 and A2, ***$p$ = 0.0007 for A3, 2 independent replicates). A1 and A2; areas of the dermal ECM, A3; areas under and inside the epidermal layer, A4; area of the deep epidermal layer. **j, k** Image-based quantification of the epidermal layer differentiation in HEK only or in the HEK + SN group. Representative immunofluorescence images (**j**) of K14/K10 layer development. Y = 0 (μm): Interface of collagen gel channel and HEK channel, Y > 0: apical (ALI), and Y < 0: basal (gel) directions (**k**) (4 independent replicates). Scale bars; 50 μm. **l, m** Epidermal layer permeability of 376.27 Da FITC-sodium at 120 min (**m**) and a 3D confocal image (**l**) of K14 (red) and K10 (green) in the epidermal layer. Scale bars; 50 μm ($n$ = 3 devices, 1 independent replicate). Data are mean ± SD, *$p$ < 0.05, **$p$ < 0.01, ***$p$ < 0.001, ****$p$ < 0.0001. Two-tailed $t$-test, two-tailed Mann–Whitney test or one-way ANOVA, Tukey's multiple comparisons test.

associated with keratinocytes and free nerve endings makes it hard to selectively stimulate keratinocytes while ignoring SN in conventional models. However, the developed innervated epidermal-like layer in a microfluidic chip allows us to overcome the pitfalls and keratinocytes to initiate nociceptive transduction, thanks to the spatial compartmentalization of the cells and ECM hydrogels with apparent controllability and perfect barrier function. The developed chip can experimentally model a hyperglycemic environment to understand pathological roles and changes of innervated skin components in the development of diabetic neuropathy. Consistent with known pathogenesis of diabetes and diabetic complications, acute hyperglycemia-induced loss of TRPV1$^+$ intraepidermal nerve fibers and disrupted development of epidermal layer by ROS accumulation rather than apoptosis. Hyperglycemia-induced impaired barrier function of epidermal-like layer penetrating topical substances suggests the reason why the skin of diabetic patients is more susceptible to barrier defects caused by external stimuli. The afferent transmission provides a possible mechanism for early features of acute hyperglycemia/prediabetes without electrophysiological evidence of nerve damage. It also could explain sensory dysfunction such as neuropathic pain behavior in diabetic patients despite the loss of intraepidermal nerve fibers.

However major limitations still remain and impede the completeness of the developed chip and protocol. Due to restricted access to primary adult human SN, the developed model utilized rodent sensory neurons (DRGs), disregarding the possible existence of interspecies differences. Recent hiPSC or hiNSC-derived SN are new translational alternatives but still remain with inconsistencies in cellular function and population compared to native human or rodent cells[33,58]. The developed protocol could be a good guide to exploit a chip with an innervated epidermal-like layer with full human origin cells for future study, because it already shows its capability for studying neurocutaneous diseases and drug screening especially for topical applications with a level of complexity not found in conventional skin models and close prediction of in vivo results by the barrier function and transmission of sensory stimuli. Our data shows a structurally and functionally integrated innervated epidermal-like layer, recapitulating abnormal cellular interactions in a pathophysiologically relevant human setting. We hope to provide insights on the future integration of these skin models with other cell components onto microfluidic platforms as well as potential readout technologies for high-throughput drug screening. We also hope to prolong the culture period to allow keratinocytes to reach a higher level of maturity, reaching the epidermis layer with IB4$^+$ nerve fibers deeply localized in the granular layer.

## Methods

### Ethical statement
The animal experiments that isolating DRG of embryonic day 15 Sprague-Dawley rat embryos were approved by the Korea University Institutional Animal Care and Use Committee (KUIACUC-2017-138).

### Microfluidic design and fabrication
The design of the chip was modified and fabricated from the previous report (Fig. 1a)[32,42]. Briefly, the device was produced from a SU-8 patterned wafer via soft lithography using polydimethylsiloxane (PDMS, SYLGARD 184; Dow Chemical Company). A PDMS replica was punched with a dermal biopsy punch (diameters of 4 mm or 1 mm, for the medium or gel channels, respectively). It was then sterilized twice at 120 °C for 20 min, followed by drying at 80 °C in an oven for 6 h. The device and a glass coverslip (18 × 18 mm$^2$, Paul Marienfeld GmbH & Co.) were bonded using oxygen plasma (CUTE; Femto Science Inc.). Then, the microchannels were filled with a 2-mg/mL polydopamine solution (PDA, Sigma-Aldrich) and maintained at room temperature for 2 h with light protection. After washing with sterilized deionized water, the devices were dried at 80 °C in an oven for 24 h and stored at room temperature for use (Supplementary Fig. 2a).

### Gel-filling procedure
Two gel channels in the microfluidic device were filled with two types of hydrogels (Fig. 1a and Supplementary Fig. 2b). First, type I collagen gel solution (Corning Inc.) was prepared by mixing 10× phosphate-buffered saline (PBS) with phenol red, 0.5 N NaOH, and deionized water to a concentration of 3 mg/mL and a pH of 7.4. This collagen gel solution was injected into the HEK-side gel channel, then the device was placed upside down in a pre-warmed humid chamber and incubated for 30 min at 37 °C in a 5% CO$_2$ atmosphere for gelation. Next, a 1.5 mg/mL collagen solution containing 10% laminin (Sigma-Aldrich) was prepared according to the previously described method. Then, it was filled in the SN-side gel channel to be connected to the pre-formed collagen channel. After gelation, the device's medium reservoir was filled with SN medium and stored in an incubator for cell seeding (Supplementary Fig. 2a).

### Cell culture
Primary SN were isolated from DRG of embryonic day 15 Sprague-Dawley rat embryos (KOATECH, Gyeonggi, South Korea) and cultured in neurobasal media (Gibco) supplemented with 250 ng/mL recombinant nerve growth factor 7 S (Sigma-Aldrich), 10% fetal bovine serum, 2% B-27 supplement, 2 mM L-glutamine, and 1% antibiotic solution (all from Gibco) for growth in a humidified 5% CO2 incubator as previously described[29,30]. For keratinocytes, Adult normal HEKs were commercially obtained from Lonza Group AG (Basel, Switzerland) (00192627) and were cultured in KGM-Gold medium (Lonza) supplemented with the KGM-Gold Bullet Kit (Lonza) according to the manufacturer's instructions.

### Co-culture of Keratinocytes and DRG neurons in the microfluidic chip
After adding 60 μL of a rat SN suspension with a density of 1.8 × 10$^6$ cells/mL to the SN channel, the device was placed vertically in an

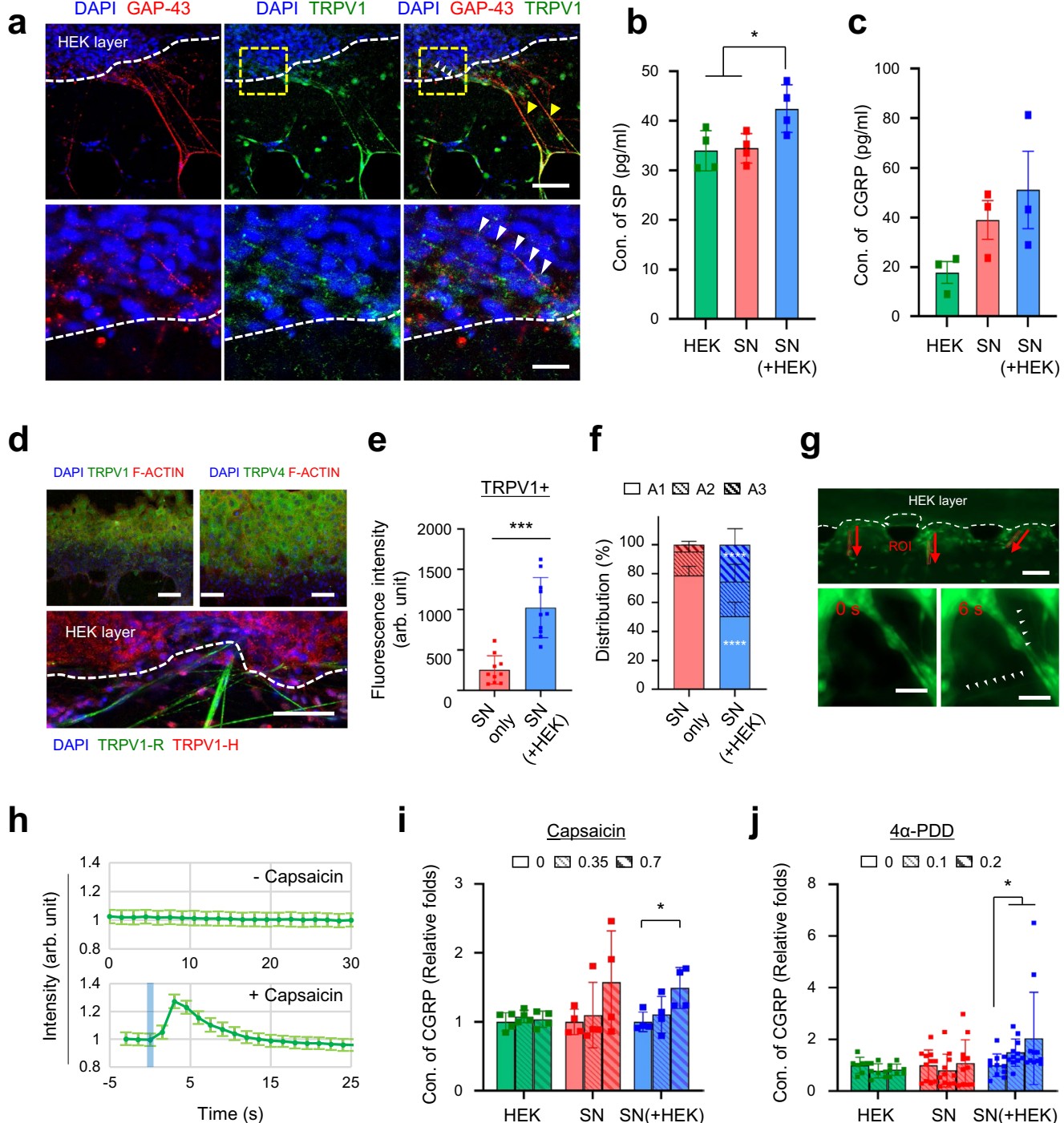

**Fig. 5 | Functional integrity of the innervated epidermal-like layer in the microfluidic chip. a** Representative immunofluorescence images of TRPV1 (green) and GAP-43 (red) expression in sensory neurons co-cultured with keratinocytes on a chip. Arrowheads indicate TRPV1+ cells co-stained with GAP-43 in either the outer epidermal and ECM layers (yellow) or the intraepidermal layer (white). White dashed line; the outer epidermal layer. Magnifications (bottom) of the region are highlighted in the yellow dashed box (top). Scale bars; 100 μm, 25 μm, respectively (1 independent replicate). **b, c** Quantification of neuropeptides released from HEK only, SN only, and SN + HEK group under unstimulated conditions. The concentration of substance P ($n = 4$ devices, SN + HEK vs HEK *$p = 0.036$, SN + HEK vs SN *$p = 0.0248$, 2 independent replicates) (**b**) or CGRP ($n = 3$ independent replicates, mean ± SEM) (**c**) is determined in culture supernatants. **d–f** TRPV1 and TRPV4 expression in the innervated epidermal chip. Representative immunofluorescence images (top of **d**) of epidermal keratinocytes TRPV1 or TRPV4 (green) and F-ACTIN (red) expression. TRPV1 expression (bottom of **d**) was confirmed with a human-specific antibody (TRPV1-H, red) or

with a rat-specific antibody (TRPV1-R, green). Scale bars; 100 μm, 50 μm, respectively. Quantification of total TRPV1+ neurons (**e**) and spatial distribution (**f**) of TRPV1+ neurons along the regions (presented in Fig. 4g) ($n = 10$ ROIs, 2 ROIs per device, SN + HEK vs SN ****$p < 0.0001$ for A1 and A3, 2 independent replicates). **g, h** Capsaicin-evoked Ca²⁺ transients of innervating sensory neurons. Intracellular Ca²⁺ images (**g**) of neurons responding to topical application of capsaicin (0.1 mM) and the fluorescence intensity time course (**h**) of peak Ca²⁺ transients (calcium fluorescence intensities along the axon was indicated mean ± SD, 2 independent replicates). **i, j** The CGRP release from sensory neurons co-cultured with keratinocytes following topical application of capsaicin (**i**, agonist for TRPV1) ($n = 4$ devices, cap(0.7) vs cap(0) *$p = 0.0286$ for SN + HEK, 2 independent replicates) or 4α-PDD (**j**, agonist for TRPV4) ($n = 7$–11 devices, cap(0.1) vs cap(0) *$p = 0.028$, cap(0.2) vs cap(0) *$p = 0.0192$ for SN + HEK, 2 independent replicates) at indicated concentrations (unit: mM). Data are mean ± SD, *$p < 0.05$, ***$p < 0.001$, ****$p < 0.0001$. Two-tailed $t$-test or two-tailed Mann–Whitney test.

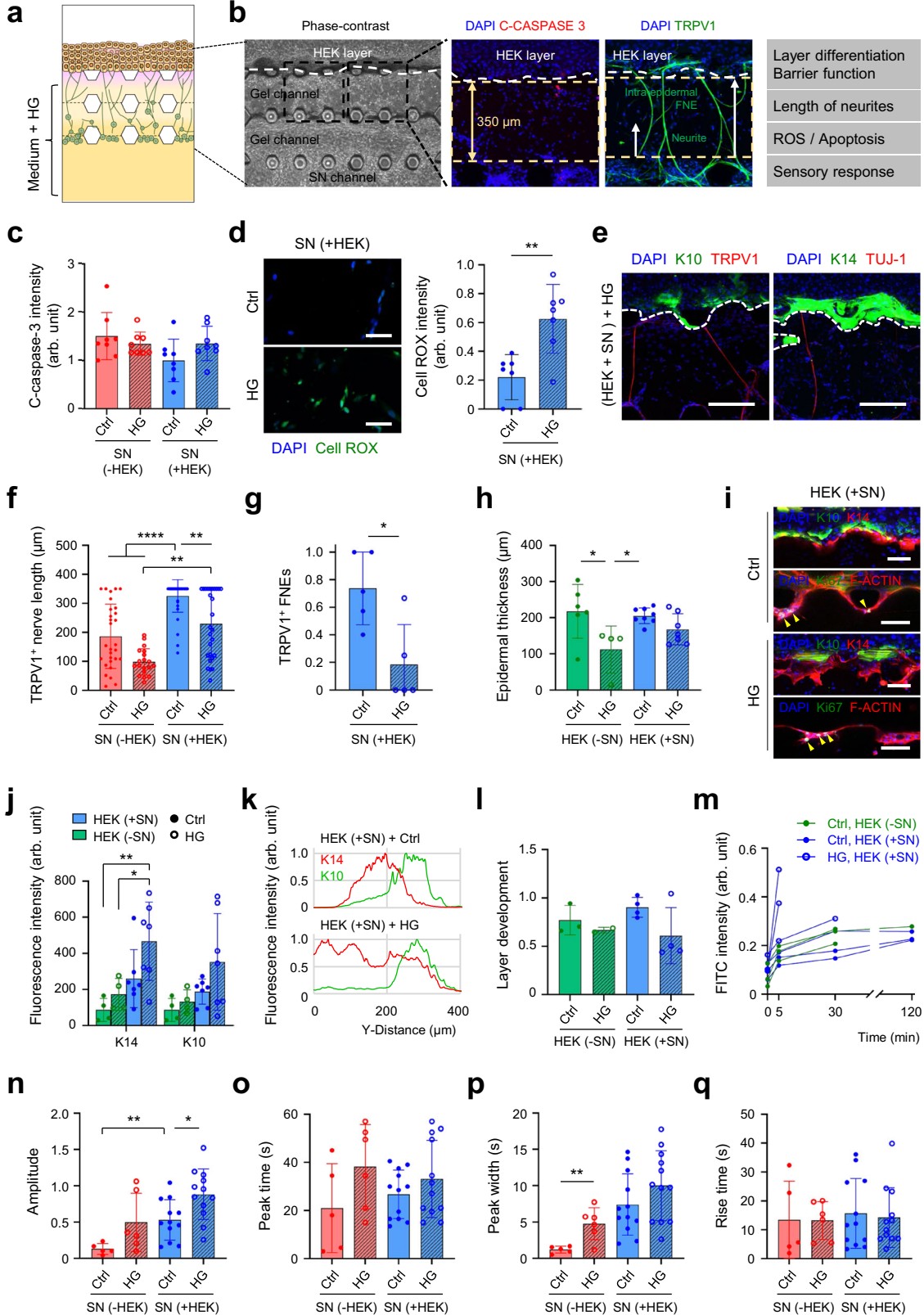

incubator for 3 h to attach the cells to the hydrogel, followed by culture for 5 days, (Fig. 1d). For co-culture of seeded SNs and keratinocytes, 60 µL of HEK cell suspension ($1.2 \times 10^6$ cells/mL) was added to the opposite channel for seeding and the device was tilted in the opposite direction for 2 h. Two days after HEK inoculation, the HEK medium was removed from the reservoir to manipulate the ALI culture conditions by placing the device at a 30° angle. The culture

medium of each cell was changed according to the culture stage; FBS-depleted neurobasal medium for SNs, ascorbic acid (50-µg/mL, Sigma-Aldrich)-added differentiation medium for HEKs as optimized and described in Fig. 3. For hyperglycemic conditions to mimic the diabetic environment, SN channels were maintained with 100 mM D-glucose (Sigma, G7021) without insulin during the slope-based culture (3 days) [43,54,55].

**Fig. 6 | Acute hyperglycemia-induced pathological modeling using innervated epidermal-like layer chips. a** Modeling of hyperglycemia (HG)-induced innervated epidermis on a microfluidic chip, and analyzing in a cell-type-specific manner (**b**). **c** Quantification of fluorescence intensity of the cleaved caspase 3$^+$ population in sensory neurons ($n = 8$ ROIs, 2 ROIs per device, Ctrl vs HG $p = 0.8536$ for SN-HEK and $p = 0.2947$ for SN + HEK, SN + HEK vs SN-HEK $p = 0.0694$ for Ctrl, 2 independent replicates). **d** Intracellular reactive oxygen species (ROS) levels in the innervating neurons ($n = 7$ ROIs, 2 ROIs per device **$p = 0.0027$, 1 independent replicates). Scale bars; 50 µm. **e** Immunofluorescence images of innervated epidermis for K14 or K10 (green) and TRPV1 or TUJ1 (red) after 3 d of high glucose exposure (2 independent replicates). Scale bars; 200 µm. **f,g** Hyperglycemia-induced changes in TRPV1$^+$ neurons are determined by quantification of neurite length (**f**) of TRPV1$^+$ neurons ($n = 19-37$ ROIs, SN + HEK (Ctrl) vs SN-HEK (Ctrl, HG) ****$p < 0.0001$, SN + HEK (Ctrl) vs SN + HEK (HG) **$p = 0.0062$, SN + HEK (HG) vs SN-HEK (HG) **$p = 0.0018$, 2 independent replicates, Kruskal–Wallis test) and free nerve endings (FNEs, **g**) of TRPV1$^+$ neurons innervating the epidermal keratinocyte layer ($n = 4-5$ devices, *$p = 0.0317$, 2 independent replicates). **h–l** Hyperglycemia-induced changes of epidermal layer development. Quantification of the epidermal thickness ($n = 4-8$ devices, HEK-SN (Ctrl) vs HEK-SN (HG) *$p = 0.0207$, HEK + SN (Ctrl) vs HEK-SN (HG) *$p = 0.0336$, 2 independent replicates) (**h**) and K14$^+$ and K10$^+$ layers (**j**) between controls and HG groups. Immunofluorescence images (**i**) of K14, K10, and ki67-positive cells (yellow arrowheads) and fluorescence intensity plots (**k**) of K14 and K10 in epidermal layers. Scale bars; 200 µm. The relative ratio of K10 over the K14 layer along the Y-axis showing layer organization (**l**) ($n = 2-4$ devices, 2 independent replicates). **m** Hyperglycemia-induced changes in epidermal permeability of 376.27 Da FITC-sodium. **n–q** Capsaicin(0.1 mM)-evoked Ca$^{2+}$ transients between controls and HG groups. Amplitude (SN + HEK (Ctrl) vs SN-HEK (Ctrl) **$p = 0.0072$, SN + HEK (Ctrl) vs SN + HEK (HG) *$p = 0.0117$) (**n**), peak time (**o**), peak width (**p**), and rise time (SN-HEK (Ctrl) vs SN-HEK (HG) **$p = 0.0067$) (**q**) ($n = 5-6$ ROIs for SN-HEK, 12 ROIs for SN + HEK, 2 ROIs per device, 2 independent replicates). Data are mean ± SD, *$p < 0.05$, **$p < 0.01$, ***$p < 0.001$, ****$p < 0.0001$. Two-tailed $t$-test, two-tailed Mann–Whitney test or one-way ANOVA, Tukey's multiple comparisons test.

## Immunofluorescence microscopy analysis

Chips were fixed with 4% paraformaldehyde in PBS for 30 min and then washed with PBS. Immunostaining was performed after permeabilization in PBS with 0.1% Triton X-100 and blocking in 3% BSA (Thermo Fisher Scientific). Antibodies used in this study were listed in Supplemental Table 2, which were incubated overnight on chips at 4 °C. Fluorescent conjugated secondary antibodies were then used and Nuclei were counterstained with 4,6-diamidino-2-phenylindole dihydrochloride (DAPI, Invitrogen). Confocal imaging was obtained using Confocal Microscope (Olympus, Japan) (LSM 700, Carl Zeiss, Germany) and analyzed using ImageJ software (https://imagej.nih.gov/ij/index.html), FluoVIEW (Olympus, Japan) or ZEN 2.3 software (Carl Zeiss, Germany).

## Quantitative real-time polymerase chain reaction (qRT-PCR) analysis

RNA was extracted from normal HEK in the microfluidic chip using TRIzol reagent (Invitrogen). Complementary DNA (cDNA) was synthesized by reverse transcription with a high-capacity RNA-to-cDNA kit (Applied Biosystems). qRT-PCR was performed using Power SYBR Green PCR Master Mix (Applied Biosystems) in the StepOne Real-time PCR equipment (Applied Biosystems). The primer sequences are listed in table S2. The gene expression levels were normalized to the housekeeping gene Gapdh and quantified with the comparative Ct method.

## Calcium imaging

Calcium imaging was performed using the Fluo-4 Direct Calcium Assay Kit according to the manufacturer's instructions (Invitrogen). Briefly, chips were incubated with a 1:1 mixture of 2× reagent and SN serum-free medium in a humidified 5% CO$_2$ incubator for 60 min and then washed with serum-free medium. Images were captured every 1500 ms for 2 min with or without 0.1 mM capsaicin treatment in the HEK channel. For determination of threshold for activation in calcium imaging, the magnitude of the ratio change during exposure to agonist (ΔF) was normalized to the baseline ratio for each imaged neuron (ΔF/F0 ratio, fold-change). A histogram of the fold-change of each individual neuron in a particular experiment was constructed for each stimulus to determine the threshold for activation. The multimodal histogram contained one large peak around 1 (defined as the background response, F0) and neurons with fold-changes greater than the first minimum (defined as threshold) were considered responsive [50,59].

## ELISA

HEK channels were pre-stimulated with capsaicin (Sigma-Aldrich), 4α-PDD (Sigma-Aldrich), or DMSO and then supernatants were harvested from the SNs channels after 2 h of agonist treatment. The concentration of CGRP in the culture supernatant was measured using a rat CGRP Enzyme Immunoassay kit (SPI-Bio) and VICTOR X3 (PerkinElmer, USA) at a wavelength of 405 nm following the manufacturer's recommended procedures.

## Reactive oxygen species (ROS) measurements

ROS was quantitatively analyzed by a fluorescent probe (Cell ROX Green Reagent, Invitrogen) for measuring oxidative stress in living cells. 5 µM Cell ROX green reagent was added to the culture medium at 37 °C for 30 min, washed with PBS and fixed with 4% PFA. After counterstaining with DAPI, fluorescence images were acquired using a confocal laser scanning microscope (LSM 700, Carl Zeiss, Germany) and analyzed with ImageJ software [49,51].

## Epidermal permeability assay

3.839 kDa FITC-dextran (Sigma, FD4) or 376.27 Da FITC-sodium (Sigma, F6377) were added to keratinocyte channels. Time-lapse epidermal permeability is determined as the flux of fluorescent tracers across the epidermal layer to ECM layer by the concentration difference for 120 min. The fluorescence intensity was analyzed and calculated using ImageJ according to the previously described method [42].

## Statistics and reproducibility

The statistical calculations of the results were carried out by Prism software (GraphPad Software, San Jose, CA, USA), and data were expressed as mean ± standard derivations (SD) for $n \geq 3$ or as mean ± SEM with at least three independent replicates. The unpaired, two-tailed Student's $t$ tests or two-tailed Mann–Whitney test was used to determine the significance of the data between the two groups. Multigroup analyses were made by one-way analysis of variance (ANOVA) followed by a Tukey's multiple comparisons test or a Bonferroni's multiple comparisons test and $p$ values below 0.05 were deemed statistically important: *$p < 0.05$, **$p < 0.01$, and ***$p < 0.001$, ****$p < 0.0001$. No statistical method was used to predetermine the sample size. Throughout the study, the sample size was determined based on our preliminary studies and on the criteria in the field.

## Reporting summary

Further information on research design is available in the Nature Portfolio Reporting Summary linked to this article.

## Data availability

All data needed to evaluate the conclusions in the paper are presented in the paper and/or the Supplementary information. Raw data for all figures are provided in Source Data file. Source data are provided with this paper.

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

## Acknowledgements

This work was supported by Samsung Research Funding & Incubation Center of Samsung Electronics under Project Number SRFC-IT1901-51 and by the Korea Evaluation Institute of Industrial Technology (KEIT) grant funded by the Korea government (MSIT) (No. 20009125). JA Kim was supported by the Technology Innovation Program (or Industrial Strategic Technology Development Program (20015148, Development of Neural/Vascular/Muscular-Specific Peptides-conjugated Bioink and Volumetric Muscle Tissue) funded By the Ministry of Trade, Industry & Energy (MOTIE, Korea).

## Author contributions

J.A. and K.O. equally designed, performed, and analyzed the experiments and wrote the paper. J.W., D.-H.C., Y.H.J., J.H.Y., and Y.J. assisted in performing experiments and generating data. J.-A.K. and S.C. designed conceptual ideas and research, supervised the study, and wrote the paper. S.C. provided financial support.

## Competing interests

The authors declare no competing interests.
