## [Peer Review File · Nature Communications]

Reviewers' Comments:

Reviewer #1:

Remarks to the Author:

The manuscript by Ohk et al. describes an in vitro "skin-nerve" model based on co-cultures of human keratinocytes and rat dorsal root ganglion (DRG) sensory neurons in microfluidic devices in a three-dimensional configuration. The DRG cells and the keratinocytes are separated by two layers of collagen hydrogels in a configuration that allows the neurons to extend their axons across the collagen layer to reach the keratinocyte compartment. The keratinocytes have been differentiated through air exposure to mimic epidermis formation in the "skin" compartment. The authors show that the DRG neurites can extend into the differentiated "epidermis" and release CGRP in response to stimulation of the keratinocyte compartment with capsaicin or phorbol ester. The authors conclude that the co-culture system described can be used for disease modelling and toxicity testing.

From a technical point of view, the methodology described in the paper is very interesting. However, my major concern is that a more in-depth characterisation of the model would be required to support the conclusions drawn in the paper, particularly in relation to the utility of the described co-culture platform for disease modeling.

specific comments:

1- It would be important to know the extent of the fluidic isolation (i.e. permeability of the hydrogel layers) between the DRG and keratinocyte compartments in the PDMS devices. This issue has not been addressed in the manuscript. Can substances such as CGRP or capsaicin permeate through the collagen hydrogels between the compartments?

2- To demonstrate the 3D outgrowth of the neurites into the collagen hydrogels, a set of 3D Z-stack projections or 3D reconstruction of the fasciculated neurites in the collagen hydrogels would be beneficial. The current images do not give any indication of the 3D outgrowth of the neurites into the hydrogel layers.

3- It is not clear if non-peptidergic sensory neurites can also enter the hydrogel layers and innervate the keratinocyte compartment. Given the functional heterogeneity of DRG sensory neurons, it would be of interest to characterise the neurites crossing into the keratinocyte compartment using markers including NF200, IB4, CGRP and TRPV1 to ascertain representation of different neuronal cell types in the co-cultures.

4- Figure 4 shows a significant increase in the basal levels of CGRP release in the co-cultures compared to single cultures of sensory neurons (figures 4f and supplementary 5b). However, the capsaicin-evoked CGRP release appears to be higher in sensory neuron cultures alone (ratios given in Figure 4f). The authors concluded that "the integrity of the innervating sensory neurites is preserved" in the co-cultures, however, this could suggest a reduced sensitivity to capsaicin or decreased CGRP content in DRG neurons maintained in the co-cultures.

5- Statistical analysis of the CGRP release ratios is not discussed in the manuscript. The p-values for statistically non-significant data should also be provided. It would be more informative to present the actual capsaicin or 4alphaPDD evoked CGRP release levels for comparison.

6- It is not clear if the CGRP release ratios are not significantly different, or that no statistical analysis of the ratios has been carried out.

7- The authors conclude that "the integrity of the innervating sensory neurites is preserved" in the co-cultures, however, whether these neurites are capable of generating and conducting action potentials to their somas, has not been addressed. Characterisation of the electrophysiological properties of the neurites in the co-cultures, using calcium imaging, for instance, would be of interest here.

8- The authors conclude based on the morphology of the neurites immunostained with PGP9.5 antibody that the neurites entering the epidermis layer "...form cutaneous nerves [endings]". However, it is not clear if these neurites terminate once entering the differentiated keratinocyte

layer or continue to extend growth cones in the "epidermis" layer. It would be important to determine if the neurites projecting through the differentiated keratinocyte layer terminate, in a similar manner to the projections in vivo, or if they continue to grow via growth cones. For instance, immunolabeling with growth cone markers such as gap-43 could be informative.

9- Figure 3 and supplementary figure 4d show the formation of the spinous layer using K10 immunostaining. Given the importance of the basal layer of the skin, it would be interesting to characterise the undifferentiated keratinocyte layer using K5 immunostaining.

10- TRPV4 has been shown to be present on sensory nerve endings and keratinocytes in the skin. Figure 4h suggests that there is an increase in the 4alphaPDD mediated release of CGRP from co-cultures of SN and keratinocytes compared to SN alone. However, the data for keratinocyte cultures alone are not presented. This would be of interest since the presence and release of CGRP from human keratinocytes has been reported (e.g Pain. 2011 Sep;152(9):2036-51). If CGRP is released from Keratinocytes in response to stimulation by 4alphaPDD, this would explain the additional CGRP release seen in the co-cultures compared to single SN cultures.

11- The advantages of the 3D model described in this manuscript over other reported co-culture or microfluidic platforms (e.g. Biomed Microdevices (2017) 19: 22, Biomaterials (2017)116:48 and PLOS ONE 8 (11), e80722) are unclear. Ultimately a proof of concept experiment to demonstrate the utility of the co-culture system in disease modelling would be of interest.

12- Furthermore, the disadvantages of the model should be clearly addressed in the discussion, particularly in the view of the inter-species nature of this model which could significantly limit its utility for human disease modeling and toxicity testing.

Reviewer #2:

Remarks to the Author:

General comment:

The study describes co-culture of epidermal keratinocytes and sensory neurons in a microfluidics device and cultured at the air liquid interface to enable keratinocyte differentiation and stratification. Neuropeptide secretion by SNs and neurite migration is reported e.g. calcitonin gene-related polypeptide. The authors claim that their "observations validated the potential of the proposed model as an in vitro cutaneous nerve model for disease studies and drug toxicity tests". This is not shown in the study. No disease model and limited drug testing is performed, with no statistical significance. Therefore this claim is exaggerated. Although interesting, it remains a potential model which requires much further extensive investigation before publication in Nature Communications.

Furthermore, use of English language needs extensive correcting throughout.

Title: is misleading as the study describes only one type of skin cell, the keratinocyte. Cutaneous indicated the complexity of the skin organ. This terminology needs correcting throughout.

Abstract: needs revising according to general comment

Introduction:

What is meant by skin being the most extensive organ in the body. What about e.g. lungs, GI tract..... This statement is incorrect.

Symbols are showing as boxes, needs correcting

"development of systemic in vitro models that simulate actual skin in the fields of pharmaceuticals and cosmetics" – what is meant by systemic? This sentence is rather vague.

What is meant by "2D laminated layers" where is evidence that most models consist of this? I have never heard of this terminology.

3D Skin equivalent models have existed for 30 years without microfluidics

Porous films are generally known as transwell inserts

The text doesn't follow, if 2D works why is 3D needed?

"disease skin models to guide therapeutic, pharmaceutical, and industrial discoveries." is very generalized and cannot be applied to this study in this way. More concrete examples are needed. Since the hydrogel lacks any dermis derived cell type, even the fibroblast, the hydrogel cannot be referred to as a dermis in this model.

What is the source of the SNs?

"complicated basal and spinous layer of epidermis" why complicated, what about granular layer and stratum corneum. Static models have these, microfluidics not needed.

The term "airway" culture suggests respiratory not skin.

Results:

Fig 1 does not show any results.

Terminology in this entire section needs significantly revising: Reconstitution of epidermis on ECM hydrogel.

What is the idea between the flat and sloped culture method. The reason for air exposure is clear as literature from many studies over many years shows that this supports epidermal differentiation. Why do you need medium flow and a chip. Why not use a simple static transwell system rather than the chip?

Please supply hematoxylin & eosin histology of the epidermis in order to see layers as indicated in fig 1. Dapi staining and the weak K10 staining are not enough to support the conclusions. Also supply K5 and Ki67 in order to see the quality of the basal layer and loricrin to see quality of the granular layer.

I miss the flat air exposed condition, this would also stimulate stratification of keratinocytes.

The word "integrity" is misplaced

Fig 4: what is A1 A2 and A3. No significance is shown in fig 4 g and h indicating no effect of tested substances.

Discussion:

Why is the microfluidic chip needed. Would the same results be obtained in an easier more scalable transwell system. If so, what effect would this have on the novelty presented here when compared to the studies presented in Supplementary table 1? I miss the static air exposed model and detailed (immuno)histology of the epidermis. The model does not contain a dermis equivalent as no cells are present in the hydrogel. Many studies ranging back 30 years describe static 3D reconstructed human epidermis on fibroblast populated collagen hydrogels. What would happen if you cultured SN cells underneath such a construct. Has a similar study already been done?

Supplementary images are of very poor quality.

S4d shows only a monolayer of keratinocytes which is not even intact (Dapi) and very little K10.

These images do not support the conclusion of epidermal stratification as suggested in S4c.

S3 and S5 black / white image extremely poor quality to draw results from

Table S1 is unclear. Eg what is yellow block. What different cell types are in the different skin models, legend is unclear

Reviewer: 1

The manuscript by Ohk et al. describes an *in vitro* “skin-nerve” model based on co-cultures of human keratinocytes and rat dorsal root ganglion (DRG) sensory neurons in microfluidic devices in a three-dimensional configuration. The DRG cells and the keratinocytes are separated by two layers of collagen hydrogels in a configuration that allows the neurons to extend their axons across the collagen layer to reach the keratinocyte compartment. The keratinocytes have been differentiated through air exposure to mimic epidermis formation in the “skin” compartment. The authors show that the DRG neurites can extend into the differentiated “epidermis” and release CGRP in response to stimulation of the keratinocyte compartment with capsaicin or phorbol ester. The authors conclude that the co-culture system described can be used for disease modelling and toxicity testing.

From a technical point of view, the methodology described in the paper is very interesting. However, my major concern is that a more in-depth characterisation of the model would be required to support the conclusions drawn in the paper, particularly in relation to the utility of the described co-culture platform for disease modeling.

1. It would be important to know the extent of the fluidic isolation (i.e. permeability of the hydrogel layers) between the DRG and keratinocyte compartments in the PDMS devices. This issue has not been addressed in the manuscript. Can substances such as CGRP or capsaicin permeate through the collagen hydrogels between the compartments?

(Answer) We deeply appreciate the reviewer's comment. Fluidic isolation by the skin layer is important to trust the signal transduction from keratinocytes to DRGs. As the reviewer pointed, the manuscript should prove barrier function of the skin layer for the smallest molecule involved in the experiments. The molecular weight of CGRP is known to about 3.789 kDa (human) and that of Capsaicin is about 33.5 kDa. The authors conducted permeability test using FITC-conjugated dextran, which has a molecular weight of 3.984 kDa.

In brief, 90 μ l of serum-free media was first filled in both channels. Under the fluorescent microscope, 10 μ l of 250 μ M FITC-dextran solution was added to the HEK channel and 10 μ l of serum-free media was to SN channel at the same time. Diffusion of the FITC-dextran was monitored by time-lapse images. The FITC-dextran signal was successfully blocked by the intact keratinocyte barrier of three devices for 2 hours, not diffusing into the ECM hydrogel. The authors added fig. 3 f-g and supplementary fig. 6.

(added figures)

Fig. 3f – g

Figure 3 | Formation of the basal and spinous layer. ... (f) Bright field and the fluorescent images captured in 2 hours after CGRP mimicking FITC-dextran (3.984 kDa) loading into the HEK channel. (g) The normalized fluorescent intensity measured from the HEK layer to ECM hydrogel (dashed line in Fig.3f).

Supplementary Fig. 6

Supplementary Figure 6. Diffusion of CGRP mimicking FITC-dextran (3.984 kDa). (a) Fluorescent intensity of the FITC-dextran applied on the leaked HEK layer and intact HEK layer. (b) Images of fluorescent signal from HEK channel to ECM, captured for 2 hours. Scale bars represent 200 μm .

2. To demonstrate the 3D outgrowth of the neurites into the collagen hydrogels, a set of 3D Z-stack projections or 3D reconstruction of the fasciculated neurites in the collagen hydrogels would be beneficial. The current images do not give any indication of the 3D outgrowth of the neurites into the hydrogel layers.

(Answer) Thank you for the suggestion. We added 3D images of three-dimensionally reconstructed neurites (stained by Tuj-1 and NF-M) near the HEK layer (stained by K10 and K14) (Fig. 1d). They show the 3D outgrowth of the neurites into the hydrogel toward the HEK layer.

(added figures)

Fig. 1d

Figure 1 | Concept of replicating the cutaneous nerve on the microfluidic platform. ... (d) 3D images of innervating neurites into the ECM hydrogel toward HEK layer. Scale bars represent 5 mm (c, left), 1 mm (c, right, top), 250 μ m (c, right, bottom) and 100 μ m (d).

3. It is not clear if non-peptidergic sensory neurites can also enter the hydrogel layers and innervate the keratinocyte compartment. Given the functional heterogeneity of DRG sensory neurons, it would be of interest to characterise the neurites crossing into the keratinocyte compartment using markers including NF200, IB4, CGRP and TRPV1 to ascertain representation of different neuronal cell types in the co-cultures.

(Answer) We appreciate the helpful comments and suggestions. To identify the types of the sensory neurites, we immunostained DRGs by NF200 (against myelinated ($A\beta$, $A\delta$) fibers, TRPV1 (against small diameter C fibers), IB4 (against non-peptidergic C fibers) and CGRP (against peptidergic C fibers) [1, 2]), as the reviewer commented. The images were added as fig. 4e and supplementary fig. 8a.

In the ECM hydrogel far from HEK layer (A1 in Fig. 4b), large number of TRPV1 neurites and double positive neurite by TRPV1 and NF200 were found. However near the HEK layer (A2 in Fig. 4b), only TRPV1 neurites were found, without NF200 expressing ones. In ECM hydrogel near HEK layer, neurites expressing IB4 or CGRP were found. However only CGRP positive neurites projected inside the HEK layer (Fig. 4e and Supplementary Fig. 8a).

Various subtypes of 3D sensory neurons in ECM hydrogel were found in the developed co-culture platform, as summarized;

- 1) far from HEK layer : myelinated C-fibers (TRPV1 and NF200)
- 2) near HEK layer : peptidergic or non-peptidergic C-fibers (CGRP or IB4 with TRPV1)
- 3) Inside HEK layer : only peptidergic C fibers (CGRP)

Fig. 4b & 4e

(added figures)

Figure 4 | Cutaneous nerve-on-a-chip reconstituting cutaneous nerve bundles. (b) the total ECM channel. The cells were stained for the cutaneous nerve (green), F-actin (red), and nuclei (blue) using PGP 9.5, rhodamine-phalloidin, and DAPI. (e) Immunofluorescence images of NF200, TRPV1, IB4 and CGRP. Scale bars represent 100 μ m in (a), (b), and (e, left) and 50 μ m in (e, right and f).

Supplementary Fig. 8a

Supplementary Figure 8. Immunostaining of sensory neurites in co-culture system. (a) NF200 and TRPV1 double positive neurites shown in the lower part of ECM channel (far from HEK layer on top), and TRPV1 positive neurites below the HEK layer. CGRP and IB4 double positive neurites beneath the HEK layer, with tiny neurites expressing CGRP protruding into the HEK layer. Scale bars represent 200 μm in (a, top) 50 μm in (a, bottom and b).

4. Figure 4 shows a significant increase in the basal levels of CGRP release in the co-cultures compared to single cultures of sensory neurons (figures 4f and supplementary 5b). However, the capsaicin-evoked CGRP release appears to be higher in sensory neuron cultures alone (ratios given in Figure 4f). The authors concluded that “the integrity of the innervating sensory neurites is preserved” in the co-cultures, however, this could suggest a reduced sensitivity to capsaicin or decreased CGRP content in DRG neurons maintained in the co-cultures.

(Answer) We certainly agree with the reviewer and appreciate the comments. The graphs of the previous manuscript were vague, misleading readers. We performed additional experiments and revised manuscript and figures, to draw clearer conclusion than before.

1) The basal level of CGRP in HEK-SN co-culture was approximately the summation of CGRP in HEK mono-culture and in SN mono-culture (Fig. 5d, left). The developed platform seemed to preserve the CGRP contents in HEK-SN co-culture.

2) Capsaicin and 4 α -PDD applied on the HEK layer could not permeate through the HEK layer, as proved in Fig. 3f-g.

3) CGRP contents measured in the medium collected from the SN channel did not correlate to the concentration of the applied stimulus (capsaicin or 4 α -PDD) on the HEK layer. However when the stimulus applied to the HEK layer with SN, the measured concentrations were proportional to the concentrations of the applied stimulus on the HEK layer (Fig. 5d, center and right, supplementary fig. 5)

4) Due to the increase of the basal level of CGRP in HEK-SN co-culture, the absolute CGRP levels in HEK-SN co-culture cases were much higher than those in HEK mono-culture cases (marked by ### in Fig. 5d).

5) CGRP contents in SN mono-culture cases were unstable, due to the reduced viability of SNs by the stimulus. Without HEK layer, neurons quickly lose their morphology by the directly applied capsaicin and 4 α -PDD, while the neurons in co-culture still maintained their network in one day after the 4 α -PDD treatment (Supplementary fig. 11).

In conclusion, the applied stimulator on the HEK layer did not directly reach the neurons. Without SN, the applied stimulator did not affect the CGRP concentration. The integrity of HEK-SN can be verified by 1) the role of mediators (i.e. ATP) from keratinocytes activating surrounding DRG neurons, and 2) TRPV channels between the DRG neurites innervating HEK layer and neighboring keratinocytes [3, 4].

(revised figures)

Fig. 5d

Figure 5 | Functional analysis of the cutaneous nerve-on-a-chip. ... (d) CGRP concentration measured by ELISA in the medium collected from SN channel. Cases include HEK or SN mono-culture and HEK-SN co-culture; CGRP concentration without stimulation (left), the ratio of the CGRP concentration under capsaicin stimulation (middle), and the ratio of CGRP concentration under 4α-PDD stimulation (right). Error bars indicate standard deviation (*, #p < 0.05, **, ##p < 0.01, ***, ###p < 0.001; * for the ratio of CGRP concentration and # for the actual values of CGRP concentration; n = 11 (left), n = 4 (middle) and n = 7 (right)).

(added figures)

Supplementary Fig. 5

Supplementary Figure 5. Absolute values of measured substances in HEK-SN co-culture cases. (a) CGRP concentration under capsaicin and 4α-PDD stimulation in the co-cultured groups. (b) Substance P (SP) concentration without stimulation in mono- and co-cultured groups. Error bars indicate standard deviation (*p < 0.05, **p < 0.01, ***p < 0.001; n = 4 (a, left), n = 7 (a, right) and n = 4 (b) for each case).

Supplementary Fig. 11

Supplementary Figure 11. HEK layer protected sensory neurons from 4α-PDD treatment. (a) Phase-contrast images of HEK, SN mono- and HEK-SN co-cultures after 4α-PDD treatment. Sensory neurons(SNs) without HEK layer were shrunk after 0.1, 0.2 mM 4α-PDD treatment but not in the co-culture condition. (b) The magnified image of SNs (red box in (a)) The scale bars represent 250 μm.

5. Statistical analysis of the CGRP release ratios is not discussed in the manuscript. The p-values for statistically non-significant data should also be provided. It would be more informative to present the actual capsaicin or 4α-PDD evoked CGRP release levels for comparison.

6. It is not clear if the CGRP release ratios are not significantly different, or that no statistical analysis of the ratios has been carried out.

(Answer for question 5 and 6) We revised fig. 5d and supplementary fig. 5 by additional experiments and statistical analysis. Thank you for the comments.

7. The authors conclude that “the integrity of the innervating sensory neurites is preserved” in the co-cultures, however, whether these neurites are capable of generating and conducting action potentials to their somas, has not been addressed. Characterisation of the electrophysiological properties of the neurites in the co-cultures, using calcium imaging, for instance, would be of interest here.

(Answer) We deeply appreciate the reviewer’s suggestions and performed additional experiments to acquire calcium imaging signal in the sensory neurites in HEK-SN co-culture condition. We added fig. 5b and supplementary fig. 10.

Before capsaicin treatment, we observed spontaneous calcium flux on individual DRG neurons but could not find calcium expressing DRG neurites. However 3 seconds after treating 0.1 mM capsaicin on the HEK layer, capsaicin-evoked calcium flux was observed at the tiny DRG neurites near the HEK layer (Fig. 5b & supplementary fig. 10). The significant increase was quantitatively confirmed by normalized fluorescent intensity.

(added figures)

Fig. 5b

b

Figure 5 | Functional analysis of the cutaneous nerve-on-a-chip.

(b) Calcium flux (green) evoked in DRG neurites (white arrowheads) after 0.1 mM capsaicin treatment on HEK channel (top). Normalized fluorescent intensity before and after the treatment (bottom). Blue line indicates the treatment point.

Supplementary Fig. 10

Supplementary Figure 10. Calcium signal in DRG neurons and neurites in HEK-SN co-culture condition. (a) Fluorescent signal of calcium flux in DRG neurons before treatment, and (b) after 0.1 mM capsaicin treatment on the HEK layer (at $t=0$ s). Calcium flux through the neurites was clearly observed (white arrowheads). t^* is arbitrary time point and scale bars represent $50\ \mu\text{m}$.

8. The authors conclude based on the morphology of the neurites immunostained with PGP9.5 antibody that the neurites entering the epidermis layer "...form cutaneous nerves [endings]". However, it is not clear if these neurites terminate once entering the differentiated keratinocyte layer or continue to extend growth cones in the "epidermis" layer. It would be important to determine if the neurites projecting through the differentiated keratinocyte layer terminate, in a similar manner to the projections in vivo, or if they continue to grow via growth cones. For instance, immunolabeling with growth cone markers such as gap-43 could be informative.

(Answer) The reviewer was certainly right. We agreed with the suggestion and immune-stained the DRG neurites entering the epidermis layer with gap-43 (against growth cone) and TRPV1 (against free nerve ending) antibodies [5,6]. Supplementary fig. 8 was added.

DRG neurons migrating into the ECM hydrogel expressed gap-43 and TRPV1. Neurites entered in HEK layer also expressed both gap-45 and TRPV1, confirming extending growth cone and formation of free nerve ending in the epidermis layer. Interestingly some neurites only expressed TRPV1 (white arrowheads). After invading HEK layer, DRG neurites seemed to lose growing capability.

(added figures)

Supplementary Fig. 8

Supplementary Figure 8. Immunostaining of sensory neurites in co-culture system.

(c) DRGs expressed TRPV1 on their soma and neurites and many of the neurites also expressed gap-43, the growth cone marker. In the HEK layer, some of neurites only expressed TRPV1, which is the marker of free nerve ending. The scale bars represent ... 100 μm in (c, top) and 25 μm in (c, bottom).

9. Figure 3 and supplementary figure 4d show the formation of the spinous layer using K10 immunostaining. Given the importance of the basal layer of the skin, it would be interesting to characterise the undifferentiated keratinocyte layer using K5 immunostaining.

(Answer) The reviewer was certainly right. We performed additional experiments for keratins by K5 / K14 (in basal) and K10 (differentiating suprabasal), and for terminally differentiated epidermal cells by Loricrin [7,8]. Fig. 4f and two supplementary figures (7b and 8b) were added.

Fig.4f show that the alignment of K10 and K14 expressing cells is much stable in HEK-SN co-culture case, K10 expressing cells above the K14 cells. When cultured with DRG neurons, polarity of the co-cultured HEK layer became stable, by reduced invasion into the ECM hydrogel and enhanced proliferation in the top layer (Supplementary fig.9). Consistent K14 and K10 alignment was noted in HEK layer co-cultured with DRG SNs (Supplementary figure 8b).

(added figures)

Fig. 4f

Figure 4 | Cutaneous nerve-on-a-chip reconstituting cutaneous nerve bundles.

(f) Immunofluorescence images of keratin in mono- or HEK-SN co-cultured HEK layers, K14 against basal layer and K10 against differentiating suprabasal. Scale bars represent 100 μm in (e, right and f).

Supplementary Fig. 7

Supplementary Figure 7. Immunofluorescent staining of HEK layers in HEK-SN co-culture condition.

K5 and K14 against basal layer, and K10 against differentiating suprabasal. Loricrin against terminally differentiated epidermal cells. Scale bars represent 100 μm .

Supplementary Fig. 8b

Supplementary Figure 8. Immunostaining of sensory neurites in co-culture system.

(b) TRPV1, Tuj-1 and NF-M expressing sensory neurites approached HEK layer, expressing K14 (against basal layer) and K10 (against differentiating suprabasal). Scale bars represent 50 μm .

10. TRPV4 has been shown to be present on sensory nerve endings and keratinocytes in the skin. Figure 4h suggests that there is an increase in the 4 α PDD mediated release of CGRP from co-cultures of SN and keratinocytes compared to SN alone. However, the data for keratinocyte cultures alone are not presented. This would be of interest since the presence and release of CGRP from human keratinocytes has been reported (e.g Pain. 2011 Sep;152(9):2036-51). If CGRP is released from Keratinocytes in response to stimulation by 4 α PDD, this would explain the additional CGRP release seen in the co-cultures compared to single SN cultures.

(Answer) We appreciated the reviewer's comment. As commented in the answer for the question 4, the fig. 5d was revised with the data of additional experiments. Presence of TRPV1 (in newly added fig. 5a) and TRPV4 (fig. 5a) in HEK layer was proved, however capsaicin and 4 α -PDD on mono-cultured HEK layer did not increase CGRP release (fig. 5d). The external stimuli in our experiments only affected DRG neurons in HEK-SN co-culture condition.

(added figures)

Fig. 5a

(revised figures)

Fig. 5d

Figure 5 | Functional analysis of the cutaneous nerve-on-a-chip. (a) Immunofluorescence images of TRPV1 and TRPV4 expression on HEK-SN cocultures. HEKs and DRG sensory neurons were double-immunostained with each TRPV1 antibody (TRPV1-DRG(D) and TRPV1-HEK(H)) ... (d) CGRP concentration measured by ELISA in the medium collected from SN channel. Cases include HEK or SN mono-culture and HEK-SN co-culture; CGRP concentration without stimulation (left), the ratio of the CGRP concentration under capsaicin stimulation (middle), and the ratio of CGRP concentration under 4 α -PDD stimulation (right). Error bars indicate standard deviation (*, #p < 0.05, **, ##p < 0.01, ***, ###p < 0.001; * for the ratio of CGRP concentration and # for the actual values of CGRP concentration; n = 11 (left), n = 4 (middle) and n = 7 (right)).

11. The advantages of the 3D model described in this manuscript over other reported co-culture or microfluidic platforms (e.g. Biomed Microdevices (2017) 19: 22, Biomaterials (2017)116:48 and PLOS ONE 8 (11), e80722) are unclear. Ultimately a proof of concept experiment to demonstrate the utility of the co-culture system in disease modelling would be of interest.

(Answer) The reviewer is certainly right. In this manuscript, the authors focused more on the realization of 3D reconstitution of cutaneous nerve in microfluidic device. Challenges and achievement of this manuscript in terms of model development are listed in supplementary table 1, carefully revised and updated during revision.

Syndromes on cutaneous nerves have complicated disease mechanism, with list of drugs and molecules. Complexity exists with unique delivery routes, stromal cells and ECMs, which should be carefully verified in the future study.

12. Furthermore, the disadvantages of the model should be clearly addressed in the discussion, particularly in the view of the inter-species nature of this model which could significantly limit its utility for human disease modeling and toxicity testing.

(Answer) We appreciated the reviewer's comment. Discussion was revised.

(revised manuscript)

Discussion

... There are several challenges of the developed microfluidic platform. First, maturity of the neurons in the epidermis-mimicking layer has not yet been fully confirmed. A previous study showed that a subset of IB4-positive non-peptidergic C-fibers terminate exclusively in the stratum granulosum of the epidermis⁴⁶, which was not clearly identified in our platform (Fig. S7-c). Only weak projections of IB4-positive non-peptidergic fibers were observed (Fig. 4e and Fig. S8-a) and free nerve endings of the peptidergic c-fibers seemed to not be fully mature, co-expressing gap-43 and TRPV1. It is known that most CGRP-positive fibers terminate in the stratum spinosum (spinous layer)⁴⁶, and the maturation of the spinous layer is required to mature peptidergic c-fiber-free nerve endings. The interspecies combination of HEKs and DRG SNs is another challenge. Utilization of human induced pluripotent stem cell (hiPSC)-derived sensory neurons⁵⁹ and Schwann cells with primary keratinocytes could address this issue, with the additional potential benefit of the development of patient-specific models.

Reviewer: 2

The study describes co-culture of epidermal keratinocytes and sensory neurons in a microfluidics device and cultured at the air liquid interface to enable keratinocyte differentiation and stratification. Neuropeptide secretion by SNs and neurite migration is reported e.g. calcitonin gene-related polypeptide. The authors claim that their “observations validated the potential of the proposed model as an in vitro cutaneous nerve model for disease studies and drug toxicity tests”. This is not shown in the study. No disease model and limited drug testing is performed, with no statistical significance. Therefore this claim is exaggerated. Although interesting, it remains a potential model which requires much further extensive investigation before publication in Nature Communications. Furthermore, use of English language needs extensive correcting throughout.

(Answer) We appreciate the reviewer’s sincere and helpful comments. The whole manuscript was carefully revised by the authors with additional experiments. After revision, language was also checked by editing service with attached certification of English editing.

Title: is misleading as the study describes only one type of skin cell, the keratinocyte. Cutaneous indicated the complexity of the skin organ. This terminology needs correcting throughout.

(Answer) We certainly agree with the reviewer. Skin includes multiple layers of epidermis and dermis, containing vascular network and nervous systems. Epidermis also bears heterogeneity, by multiple aspects of keratinocytes from non-differentiated to fully differentiated. This

manuscript deals with complicated aspects of epidermis and nervous systems with sensing capabilities. Revised manuscript includes immunofluorescent images of various phases of keratinocytes and DRG neurons, taking the initial step of 'cutaneous' nerve on a chip. We revised large part of the manuscript but hope to keep the word of 'cutaneous nerve on a chip' in title to encourage recent progress of 'tissue on a chip' field.

Abstract: needs revising according to general comment

(Answer) Not only the abstract, the whole manuscript was carefully revised.

Abstract

There is a growing need for in vitro models of skin that help accelerate the development of drugs and replacement of animal experiments. **This study describes the development of a three-dimensional epidermal nerve model in the extracellular matrix (ECM) incorporating a microfluidic chip. The proposed model co-cultured human epidermal keratinocytes (HEKs) and sensory neurons (SNs) across the ECM hydrogel. We investigated 3D neurite outgrowth into ECM hydrogel toward the epidermis-mimicking layer cultured under an air-liquid interface. The model revealed the direct integration of the epidermis and innervation with tiny neurites of SNs by the calcitonin gene-related polypeptide, and demonstrated the evoked calcium flux on tiny innervating neurites following external stimulation of the epidermis. These observations validate the potential of the proposed model as an in vitro epidermal nerve model for investigating cutaneous sensory innervation.**

Introduction:

1. What is meant by skin being the most extensive organ in the body. What about e.g. lungs, GI tract..... This statement is incorrect.

(Answer) The authors referred a challenging reference on skin, beautifully describing skin as an organ. However the authors agreed with the reviewer and corrected 'organ' by 'barrier', and added additional reference.

(revised manuscript)

Introduction

Skin is the first barrier of the human body¹ and has a role as a sensory organ for external stimuli². It contains vascular networks, peripheral nerves, and diverse appendages involved with physiological sensing for protection from external hazards³...

2. Symbols are showing as boxes, needs correcting

(Answer) It could be an issue of the pdf generation by journal submission system. Revision submission will be carefully revised. Sorry for the inconvenience.

3. “development of systemic in vitro models that simulate actual skin in the fields of pharmaceuticals and cosmetics” – what is meant by systemic? This sentence is rather vague.

4. What is meant by “2D laminated layers” where is evidence that most models consist of this? I have never heard of this terminology.

(Answer for question 3 and 4) We apologize for the confusion our terminologies have caused. The authors certainly agreed and revised the manuscript.

(revised manuscript)

Introduction

Experimental approaches for modeling the structure of skin in vitro have been studied since the 1980s^{16,17}. Profound restrictions in animal experimentation have driven the development of in vitro skin models that simulate actual skin to test topical or systemic drug actions in the fields of pharmaceuticals and cosmetics¹⁸.

5. 3D Skin equivalent models have existed for 30 years without microfluidics. Porous films are generally known as transwell inserts. The text doesn't follow, if 2D works why is 3D needed?

(Answer) The authors agreed with the reviewer's concern, and carefully arranged the introduction with more details. As the reviewer commented, 3D skin models have formed multiple layers of epidermis in transwell inserts. However microfluidic culture is beneficial by the precise 3D reconstitution with epidermis layer on the collagen rich dermal-mimicking hydrogel. 3D spatial alignment in microfluidic platform enabled co-culture of 3D DRG sensory neurons, showing peptidergic interaction with epidermis mimicking layer and exhibiting various somatosensory neuronal subtypes at the same time. 3D microfluidic platform enabled the process of this epidermal innervation and clear visualization of the reconstituted units' functions. In the revised introduction, previous references were carefully reviewed and re-arranged to present the benefit of the developed microfluidic skin platform.

(revised manuscript)

Introduction

Experimental approaches for modeling the structure of skin *in vitro* have been studied since the 1980s^{16,17}. Profound restrictions in animal experimentation have driven the development of *in vitro* skin models that simulate actual skin to test topical or systemic drug actions in the fields of pharmaceuticals and cosmetics¹⁸. As human skin equivalent (HSE) models have been developed in transwells, it has become possible to study the function of the skin including its barrier properties by mimicking the epidermal layer⁶⁴. However, the complicated structure of skin, with its various appendages, has been a major challenge to the implementation of

complicated and accurate skin models. Some pioneering studies have cultured human epidermal keratinocytes (HEKs) and dorsal root ganglia (DRG) sensory neurons (SNs) from various species (e.g., rat, porcine, and mouse) together^{60, 62, 63}, and showed *in vitro* interactions between HEKs and DRG SNs by APT-mediated signal transfer from keratinocytes to DRG SNs^{10,61,63}. An increase in the growth of keratinocytes by neuropeptide signaling from DRG SNs⁶⁵ or elevated intracellular calcium in DRG SNs by stimulation of the epidermal layer has also been presented⁶⁶. These studies have enhanced the complexity of the *in vitro* skin model, but only by co-culturing without direct contact or by projected DRG neurites onto the HEK layers. The studies could not replicate the complexity of innervating sensory nerves in epidermal layers. Recently reported microfluidic skin models have been successful in imitating major functions of the skin layer, particularly in blocking or transporting specific molecules^{19,20,21}. These models have provided benefit through the temporal application of drugs and also through the precise regulation of fluidic culture conditions, but fail to replicate the 3D reconstitution of innervating sensory nerves in epidermal layer, the achievement of which would improve our understanding of the mechanisms underlying skin sensation.

6. “disease skin models to guide therapeutic, pharmaceutical, and industrial discoveries.” is very generalized and cannot be applied to this study in this way. More concrete examples are needed.

(Answer) We appreciate the reviewer’s comment. We investigated the reconstruction of peptidergic sensory nerve endings and revealed the functional interactions between epidermal keratinocytes and DRG neurons in the developed microfluidic platform. With this platform, diseases can be modeled, i.e. hyper-innervation, the typical symptom in the skin of Atopic dermatitis patients by imbalance between nerve elongation and repulsion factors [9].

In this manuscript, we focused more on the realization of 3D reconstitution of cutaneous nerve in microfluidic platform. Challenges and achievement of this manuscript in terms of model development are listed in supplementary table 1, carefully revised and updated during revision. Syndromes on cutaneous nerves have complicated disease mechanism, with list of drugs and molecules. Complexity exists with unique delivery routes, stromal cells and ECMs, which should be carefully verified in the future study. In revision, we deleted the sentence.

7. Since the hydrogel lacks any dermis derived cell type, even the fibroblast, the hydrogel cannot be referred to as a dermis in this model.

(Answer) We substitute the word 'dermis' with 'dermis-mimicking' in the whole manuscript. Thank you for the comment.

8. What is the source of the SNs?

(Answer) We already described the source of SNs in Methods, but changed the word SNs into DRG SNs in revision. Apologies for the confusion.

Methods

Primary DRG isolation

Primary DRG were isolated from embryonic day 15 rat embryos (KOATECH, Gyeonggi, South Korea) via a surgical procedure...

9. “complicated basal and spinous layer of epidermis” why complicated, what about granular layer and stratum corneum. Static models have these, microfluidics not needed.

(Answer) Thank you for the comment. We performed additional experiments for keratins by K5 / K14 (in basal) and K10 (differentiating suprabasal), and for terminally differentiated epidermal cells by Loricrin [7,8]. Fig. 4f and two supplementary figures (7b and 8b) were added.

Figure 4 | Cutaneous nerve-on-a-chip reconstituting cutaneous nerve bundles.

(f) Immunofluorescence images of keratin in mono- or HEK-SN co-cultured HEK layers, K14 against basal layer and K10 against differentiating suprabasal. Scale bars represent 100 μm in (e, right and f).

Supplementary Fig. 7

Supplementary Figure 7. Immunofluorescent staining of HEK layers in HEK-SN co-culture condition.

K5 and K14 against basal layer, and K10 against differentiating suprabasal. Loricrin against terminally differentiated epidermal cells. Scale bars represent 100 μm .

Supplementary Fig. 8b

Supplementary Figure 8. Immunostaining of sensory neurites in co-culture system.

(b) TRPV1, Tuj-1 and NF-M expressing sensory neurites approached HEK layer, expressing K14 (against basal layer) and K10 (against differentiating suprabasal). Scale bars represent 50 μ m.

10. The term "airway" culture suggests respiratory not skin.

(Answer) We appreciate the comment and substituted 'airway culture condition' with 'air-liquid interface culture condition'. Definition of 'air-liquid interface culture condition' in our model was also added in revision.

11. Fig 1 does not show any results. Terminology in this entire section needs significantly revising: Reconstitution of epidermis on ECM hydrogel.

(Answer) Thank you for the valuable comment. The authors revised the figure 1 to describe the procedures and include results.

(revised figures)

Fig. 1b – 1d

Figure 1 | Concept of replicating the cutaneous nerve on the microfluidic platform. ... (b) Procedure of modeling the cutaneous nerve-on-a-chip microfluidic device by co-culturing HEKs and SNs. (c) Image of the microfluidic device (left), channel area (right, top) and the phase-contrast image of co-cultured HEKs and SNs. (d) 3D images of innervating neurites into the ECM hydrogel toward HEK layer. Scale bars represent 5 mm (c, left), 1 mm (c, right, top), 250 μ m (c, right, bottom) and 100 μ m (d).

12. What is the idea between the flat and sloped culture method. The reason for air exposure is clear as literature from many studies over many years shows that this supports epidermal differentiation.

(Answer) Thick epidermal layer prohibited leakage of medium into the air channel, and maintained air-liquid interface in the microfluidic platform both under flat and sloped conditions. As the reviewer commented, air exposure of HEK layer in transwell supports epidermal differentiation. However in transwell, heavy soluble components in medium below the HEK layer could not diffuse toward HEK layer against gravity. The sloped microfluidic model precisely controls tiny flow from SN channel toward HEK layer by the head difference between the HEK channel reservoirs and SN channel reservoirs. It enhanced communication between SNs and HEK layer, which is advantageous against transwell model.

(revised figures)

Fig. 1c

Figure 1 | Concept of replicating the cutaneous nerve on the microfluidic platform. ... (c) Image of the microfluidic device (left), channel area (right, top) and the phase-contrast image of co-cultured HEKs and SNs... Scale bars represent 5 mm (c, left), 1 mm (c, right, top), 250 μm (c, right, bottom)...

Supplementary Fig. 4

Supplementary Figure 4. Air-liquid culture system for the spinous layer. (a) Images of the microfluidic device with the flat-liquid, slope-liquid, and slope-air-liquid conditions...

13. Why do you need medium flow and a chip. Why not use a simple static transwell system rather than the chip?

(Answer) As answered above, the sloped microfluidic model precisely controls tiny flow from SN channel toward HEK layer by the medium head difference. It enhanced communication between DRG SNs and HEK layer, which is advantageous against transwell model. Secreted factors from one cell are dramatically enriched in tiny microfluidic channel, resulting effective paracrine signaling between cells.

14. Please supply heamatoxylin & eosin histology of the epidermis in order to see later as indicated in fig 1. Dapi staining and the weak K10 staining are not enough to support the conclusions. Also supply K5 and Ki67 in order to see the quality of the basal layer and loricrin to see quality of the granular layer.

(Answer) The authors appreciate the reviewer's comment, but histology data of the cells in microfluidic platform is very tricky to make. Instead we added immunostained images of HEK layers. The spinous and the basal layers were characterized by K5 / K14 (in basal) and K10 (differentiating suprabasal). Terminally differentiated epidermal cells were by Loricrin as the reviewer has commented. Figure 1d, Fig. 4f and two supplementary figures (7c and 8b) were added.

Fig.4f show that the alignment of K10 and K14 expressing cells (K10 cells above the K14 cells) in HEK layer was stable when co-cultured with DRG neurons. DRG Co-culture seemed to help the HEK layer stable, showing reduced invasion into the ECM hydrogel and enhanced

proliferation in the top layer (Supplementary fig.9). Supplementary figure 8b showed consistent K14 and K10 alignment in HEK layer co-cultured with DRG neurons.

(added figures)

Fig. 1d

Figure 1 | Concept of replicating the cutaneous nerve on the microfluidic platform. ... (d) 3D images of innervating neurites into the ECM hydrogel toward HEK layer. Scale bars represent 100 μm (d).

Fig. 4f

Figure 4 | Cutaneous nerve-on-a-chip reconstituting cutaneous nerve bundles.

(f) Immunofluorescence images of keratin in mono- or HEK-SN co-cultured HEK layers, K14 against basal layer and K10 against differentiating suprabasal. Scale bars represent 100 μm in (e, right and f).

Supplementary Fig. 7c

Supplementary Figure 7. Immunofluorescent staining of HEK layers in HEK mono-culture condition. K5 against basal layer, and K10 against differentiating suprabasal. Loricrin against terminally differentiated epidermal cells. Ki67 is the proliferation marker. Scale bars represent 100 μm .

Supplementary Fig. 8b

Supplementary Figure 8. Immunostaining of sensory neurites in co-culture system.

(b) TRPV1, Tuj-1 and NF-M expressing sensory neurites approached HEK layer, expressing K14 (against basal layer) and K10 (against differentiating suprabasal). Scale bars represent 50 μm .

Discussion

Although the K14 and K10 were expressed in the HEK layer, the loricrin expression was not separated with other layers and Ki67 was rarely expressed in this condition (Fig S7-b and S7-c).

15. I miss the flat air exposed condition, this would also stimulate stratification of keratinocytes.

(Answer) We performed additional experiments to check the role of the flat air–liquid interface condition to form epidermis-mimicking layer. We immunostained epidermis-mimicking layers cultured under sloped or flat (non sloped) air–liquid interface conditions. The K10 expression was reduced in HEK layer under the flat (non sloped) air–liquid interface conditions than under sloped one. (Supplementary figure 7). Air-liquid interface seemed to be affected by pressure head generated by the head difference between the reservoirs (60 μl medium in the SN channel reservoirs and no medium in the HEK channel reservoirs).

Results

We found that applying a slope of 30° to the device was successful in assisting keratinocyte differentiation into a spinous layer^{28,29}. However, the invasion of HEKs into the ECM hydrogel increased after day 4–6 in the sloped air–liquid interface condition (Fig. S7-a). The removal of the slopes (non sloped air–liquid interface condition) reduced the invasion but also the expression of K10 (Fig. S7-a and S7-b) at the same time. Barrier functions of these constructed epidermis-mimicking layers were evaluated by permeability tests with 3.984 kDa FITC-conjugated dextran. The applied dextran in the HEK channel was fully blocked by the intact HEK layer for 2 hours (Figs. 3f and 3g). However, if there were defects in the HEK layer, the dextran quickly diffused into the ECM hydrogels (Fig. S6).

(added figures)

Supplementary Fig. 7

Supplementary Figure 7. HEK layer formation in slope and no slope conditions. (a) Phase-contrast images of the HEK layers in mono-culture at 1, 2, 4 and 6 days after seeding. HEKs formed a layer while pushing the collagen gel under slope- or no slope-conditions until day 4. In slope condition, the epidermal layer continued to push the collagen gel until day 6, however in no slope condition, the HEKs tend to migrate upward and formed stabilized layer near the gel-media channel interface. The trace made by HEKs were observed on the collagen gel (arrowheads). (b) Immunofluorescence images of the HEK layers in mono-culture with slope and no slope conditions. The scale bars represent 250 μm in (a) and 100 μm in (b).

16. The word “integrity” is misplaced

(Answer) We performed additional experiments with structural (Fig. 5a & supplementary figure 8) and functional assessments (Fig. 5d & supplementary figure 10) to show the integrity of epidermal nerves. However we deleted the word integrity in revision, to make the manuscript clear.

Results

Without stimulation, CGRP was rarely detected in the medium of the monocultured HEK layer. Significantly increased amount of CGRP was detected in the medium of mono-cultured DRG SNs and even more in the medium of the co-cultured DRG SNs (Fig. 5d). This indicated the integrity of the cutaneous nerve bundles

(added figures)

Fig. 5a

Fig. 5b

(revised figures)

Fig. 5d

Figure 5 | Functional analysis of the cutaneous nerve-on-a-chip. (a) Immunofluorescence images of TRPV1 and TRPV4 expression on HEK-SN cocultures. HEKs and DRG sensory neurons were double-immunostained with each TRPV1 antibody (TRPV1-DRG(D) and TRPV1-HEK(H)) (b) Calcium flux (green)

evoked in DRG neurites (white arrowheads) after 0.1 mM capsaicin treatment on HEK channel (top). Normalized fluorescent intensity before and after the treatment (bottom). Blue line indicates the treatment point. ... (d) CGRP concentration measured by ELISA in the medium collected from SN channel. Cases include HEK or SN mono-culture and HEK-SN co-culture; CGRP concentration without stimulation (left), the ratio of the CGRP concentration under capsaicin stimulation (middle), and the ratio of CGRP concentration under 4 α -PDD stimulation (right). Error bars indicate standard deviation (*, #p < 0.05, **, ###p < 0.01, ***, ###p < 0.001; * for the ratio of CGRP concentration and # for the actual values of CGRP concentration; n = 11 (left), n = 4 (middle) and n = 7 (right)). Scale bars represent 100 μ m in (a, top), 50 μ m in (a, bottom) and (b).

(added figures)

Supplementary Fig. 10

Supplementary Figure 10. Calcium signal in DRG neurons and neurites in HEK-SN co-culture condition. (a) Fluorescent signal of calcium flux in DRG neurons before treatment, and (b) after 0.1 mM capsaicin treatment on the HEK layer (at t=0 s). Calcium flux through the neurites was clearly observed (white arrowheads). t* is arbitrary time point and scale bars represent 50 μ m.

17. Fig 4: what is A1 A2 and A3. No significance is shown in fig 4 g and h indicating no effect of tested substances.

(Answer) The locations of A1, A2 and A3 were described in the figures and manuscript. We apologize for the confusion our figures caused.

To confirm the significance, we performed additional experiments and revised manuscript and figures.

- 1) The basal level of CGRP in HEK-SN co-culture was approximately the summation of CGRP in HEK mono-culture and in SN mono-culture (Fig. 5d, left). The developed platform seemed to preserve the CGRP contents in HEK-SN co-culture.
- 2) Capsaicin and 4 α -PDD applied on the HEK layer could not permeate through the HEK layer, as proved in Fig. 3f-g.
- 3) CGRP contents measured in the medium collected from the SN channel did not correlate to the concentration of the applied stimulus (capsaicin or 4 α -PDD) on the HEK layer. However when the stimulus applied to the HEK layer with SN, the measured concentrations were proportional to the concentrations of the applied stimulus on the HEK layer (Fig. 5d, center and right, supplementary fig. 5)
- 4) Due to the increase of the basal level of CGRP in HEK-SN co-culture, the absolute CGRP levels in HEK-SN co-culture cases were much higher than those in HEK mono-culture cases (marked by ### in Fig. 5d).
- 5) CGRP contents in SN mono-culture cases were unstable, due to the reduced viability of SNs by the stimulus. Without HEK layer, neurons quickly lose their morphology by the directly applied capsaicin and 4 α -PDD, while the neurons in co-culture still maintained their network in one day after the 4 α -PDD treatment (Supplementary fig. 11).

In conclusion, the applied stimulator on the HEK layer did not directly reach the neurons. Without SN, the applied stimulator did not affect the CGRP concentration. The integrity of HEK-SN can be verified by 1) the role of mediators (i.e. ATP) from keratinocytes activating surrounding DRG neurons, and 2) TRPV channels between the DRG neurites innervating HEK layer and neighboring keratinocytes [3, 4]. Presence of TRPV1 (in newly added fig. 5a) and TRPV4 (fig. 5a) in HEK layer was proved, however capsaicin and 4 α -PDD on mono-cultured HEK layer did not increase CGRP release (fig. 5d). The external stimuli in our experiments only affected DRG neurons in HEK-SN co-culture condition.

Results

... Morphologies of neurite bundles were quantified near the starting position (A1, 600 μ m above the aggregated bodies of SNs) and after 800 μ m (A2) and 900 μ m (A3) of migration...

Figure 4 | Cutaneous nerve-on-a-chip reconstituting cutaneous nerve bundles. (a) Immunofluorescence images of the SN mono- and HEK-SN co-culture and (b) the total ECM channel. The cells were stained for the cutaneous nerve (green), F-actin (red), and nuclei (blue) using PGP 9.5, rhodamine-phalloidin, and DAPI. Quantification of (c) the number of neurites and (d) the neurite width of the SN mono- and HEK-SN co-culture in A1, A2, and A3 ($n = 3$).

(revised figures)

Fig. 5d

Figure 5 | Functional analysis of the cutaneous nerve-on-a-chip. ... (d) CGRP concentration measured by ELISA in the medium collected from SN channel. Cases include HEK or SN mono-culture and HEK-SN co-culture; CGRP concentration without stimulation (left), the ratio of the CGRP concentration under capsaicin stimulation (middle), and the ratio of CGRP concentration under 4 α -PDD stimulation (right). Error bars indicate standard deviation (*, # $p < 0.05$, ** , ## $p < 0.01$, *** , ### $p < 0.001$; * for the ratio of CGRP concentration and # for the actual values of CGRP concentration; $n = 11$ (left), $n = 4$ (middle) and $n = 7$ (right)).

Discussion:

18. Why is the microfluidic chip needed. Would the same results be obtained in an easier more scalable transwell system. If so, what effect would this have on the novelty presented here when compared to the studies presented in Supplementary table 1?

(Answer) As commented the introduction part, microfluidic skin culture enables cross-sectional investigation of 3D alignment of HEK layers and DRG SNs. 3D Growth of bundled DRG SNs into the dermis-mimicking hydrogel was sequentially monitored. The bundles were untangled when approaching near the HEK layers, and tiny neurites from the bundles invaded into the HEK layer. Neurite projection and distribution in the epidermis-mimicking layer could also be visualized. Capability of the clear visualization of 3D cell-cell and cell-ECM interactions is the most important advantage of microfluidic format.

It is also well known that the secreted factors from cells are enriched in microfluidic scale. Microfluidic co-culture dramatically increases efficiency of paracrine cell-cell, which helps the emergence of new features not shown in transwell platform. Supplementary table 1 showed that more markers were found in the 3D microfluidic models, possibly due to the enhanced

communication between cells by enriched components. Compared to the previous 3D co-culture models, our work successfully explored enhanced integrity of HEK layer and DRG neurons as shown in Supplementary table 1.

Discussion

A challenge is the optimization of the ECM hydrogel components in a microfluidic chip to facilitate the induction of both the active outgrowth of neurites and the well-organized epidermis-mimicking layer. The DRG SNs three-dimensionally growing into the ECM (type 1 collagen) hydrogel preferred a soft ECM hydrogel with a low concentration (COL1.5, 1.5 mg/mL) during attachment, which generated longer neurites as observed in previous studies involving DRG explants^{35,36}. The laminin component in the ECM hydrogel hardly affected the growth of neurites as expected³⁷, but did help the neurites to be arranged vertically toward the other side of the hydrogel. On the other hand, the HEKs required a stiff type 1 collagen ECM hydrogel to form a stable and thick layer. The microfluidic chip was designed to form a Janus ECM hydrogel with one side stiff (for HEKs) and the other side soft with laminin (for SNs). **The HEKs formed a stable layer on the stiff side and reconstructed terminal differentiation in sloped air-liquid interface condition^{29,38,39}.**

... The 3D process of epidermal nerve formation involving the untangling of nerve bundles near the epidermis and the penetration of tiny neurites into the epidermis was newly discovered in the developed microfluidic platform. NF200-positive neurites terminated before the epidermis-mimicking layer, and the TRPV1-expressing neurites penetrated into the epidermis-mimicking layer. A δ -fibers were known to be terminated in the dermis and the peptidergic and non-peptidergic C-fibers in different epidermal layers⁷. The neurite distribution in the developed platform was similar to that previously reported in *in vivo* studies⁴⁶.

... The integrity of the epidermal sensory nerves was further verified through calcium imaging. Calcium flux was observed to appear in the tiny neurites just below the epidermis-mimicking layer when the capsaicin was treated on the other side. Microfluidic platform is beneficial by the enhanced visualization of the signal transfer, by compartmented layers of multiple tissue types in the xy-plane²⁴.

(revised table)

Supplementary Table 1

In vitro models of Sensory-Skin interaction			2D										3D				work
Reference number			[1]	[2]	[3]	[4]	[5]	[6]	[7]	[8]	[12]	[3]	[9]	[10]	[11]		
Cell type	Peripheral sensory neurons	DRG	F-11	P	R	R	R	P	M	M	R, M	R	M	R	P	R	
	Epidermal cells	Keratinocyte	H	P	H	R, H	R, H	P	H	M	R	H	H	H	H	H	
Dermal cells	Fibroblast												H	H	H		
	Endothelial cell											H					
Protein marker	Pan-neuronal markers	PGP 9.5															
		Neurofilament															
		SMI 312															
		Peripherin															
		β-III tubulin (Tuj1)															
		NF200															
	Peripheral sensory neuron subtypes	TrkB															
		TRPV1															
		CGRP / SP															
		TrkA															
IB4																	
Differentiated layer properties (epidermis)	K1																
	K10																
	Involucrin																
basal layer properties (epidermis)	Filaggrin																
	Ki-67																
	K5																
Receptor expression (epidermis)	K14																
	P2Y ₂																
Analysis	Epidermis thickness	TRPV1															
		TRPV4															
	Neurites outgrowth	TRPV1															
		TRPV4															
	CGRP concentration	Basal level															
		TRPV1 activation	Cap														Cap
	SP concentration	TRPV4 activation															4α-PDD
		Basal level															
	Calcium imaging	TRPV1 activation	Cap														
		TRPV4 activation															
	ATP imaging	TRPV1 activation															
	Electrophysiology	TRPV1 activation															
Permeability test (Barrier function)	TRPV1 activation																

2D substrate
 3D aggregate
 HSE
 Microfluidic device

Table S1. Previous in vitro models of sensory-skin interaction. 2D models presented the interaction of epidermal keratinocytes and peripheral DRG sensory neurons (DRG SNs), however not with epidermal layer of differentiated and basal keratinocyte layers. Human skin equivalent (HSE) models have in vivo like epidermal layer under air-liquid interface culture condition, but lack the direct and sequential visualization of epidermal free nerve ending formation in 3D. F-11: F-11 cell line (a mouse N18TG2 neuroblastoma X rat DRG sensory neuron hybrid cell line), H: human, P: porcine, R: rat, M: mouse, Cap: capsaicin, 4α-PDD: 4α-phorbol 12,13-didecanoate, ATP: adenosine triphosphate, Mech: mechanical stimulation, Elec: electrical stimulation, Heat: Heat stimulation.

19. I miss the static air exposed model and detailed (immuno)histology of the epidermis.

(Answer) The epidermis-mimicking layer and epidermal nerves in our model were cultured in static condition. Sorry for the missed description, which was revised.

20. The model does not contain a dermis equivalent as no cells are present in the hydrogel. Many studies ranging back 30 years describe static 3D reconstructed human epidermis on fibroblast populated collagen hydrogels. What would happen if you cultured SN cells underneath such a construct. Has a similar study already been done?

(Answer) As the reviewer commented, cellular dermal components (i.e. fibroblast and endothelial cells) are expected to affect proliferation, differentiation and maturation of epidermis-mimicking layer and DRG sensory neurons. However in microfluidic platform, stable culture of the cellular dermal components in dermis-mimicking hydrogel is very tricky, due to the active remodeling of hydrogel by the cells. The authors hope to explore it in future study.

Supplementary Information:

21. S4d shows only a monolayer of keratinocytes which is not even intact (Dapi) and very little K10. These images do not support the conclusion of epidermal stratification as suggested in S4c.

(Answer) We revised the immunostained images of K10 to support the epidermal stratifications in our epidermis-mimicking layer (Fig. 1d & Fig. 4f). Additional images described above would also be helpful for the confirmation. Thank you for the valuable comment.

(added figures)

Fig. 1d

Figure 1 | Concept of replicating the cutaneous nerve on the microfluidic platform. ... (d) 3D images of innervating neurites into the ECM hydrogel toward HEK layer. Scale bars represent 5 mm (c, left), 1 mm (c, right, top), 250 μ m (c, right, bottom) and 100 μ m (d).

Fig. 4f

Figure 4 | Cutaneous nerve-on-a-chip reconstituting cutaneous nerve bundles.

(f) Immunofluorescence images of keratin in mono- or HEK-SN co-cultured HEK layers, K14 against basal layer and K10 against differentiating suprabasal. Scale bars represent 100 μm in (e, right and f).

22. S3 and S5 black / white image extremely poor quality to draw results from Table S1 is unclear. Eg what is yellow block. What different cell types are in the different skin models, legend is unclear

(Answer) We appreciate the reviewer's comment. The authors revised the supplementary figure 3 and supplementary table 1. Supplementary figure 5 was deleted.

(revised figures)

Supplementary Fig. 3a

Supplementary Figure 3. SNs under different gel conditions. (a) Phase-contrast images of SNs cultured in the COL2, COL2L, and COL1,5L gel conditions. The scale bars represent 100 μm .

(revised table)

Supplementary Table 1

In vitro models of Sensory-Skin interaction			2D									3D					this work
Reference number			[1]	[2]	[3]	[4]	[5]	[6]	[7]	[8]	[12]	[3]	[9]	[10]	[11]		
Cell type	Peripheral sensory neurons	DRG	F-11	P	R	R	R	P	M	M	R, M	R	M	R	P	R	
	Epidermal cells	Keratinocyte	H	P	H	R, H	R, H	P	H	M	R	H	H	H	H	H	
	Dermal cells	Fibroblast											H	H	H		
Protein marker	Pan-neuronal markers	PGP 9.5															
		Neurofilament															
		SMI 312															
		Peripherin															
		β-III tubulin (Tuj1)															
	Peripheral sensory neuron subtypes	NF200															
		TrkB															
		TRPV1															
		CGRP / SP															
		TrkA															
	Differentiated layer properties (epidermis)	IB4															
		K1															
		K10															
		Involucrin															
	basal layer properties (epidermis)	Filaggrin															
		Ki-67															
		K5															
	Receptor expression (epidermis)	K14															
		P2Y ₂															
		TRPV1															
Analysis	Epidermis thickness																
	Neurites outgrowth																
	Neurites projection to epidermis																
	CGRP concentration	Basal level															
		TRPV1 activation	Cap														Cap
		TRPV4 activation															4α-PDD
	SP concentration	Basal level	Cap														
		TRPV1 activation	Cap						Cap								
	Calcium imaging			Mech Cap	Mech	ATP Cap			Mech ATP	ATP	Cap	Mech Cap		Cap			Cap
	ATP imaging								Mech Heat								
	Electrophysiology								ATP		Elec						
	Permeability test (Barrier function)																

2D substrate
 3D aggregate
 HSE
 Microfluidic device

Table S1. Previous in vitro models of sensory-skin interaction. 2D models presented the interaction of epidermal keratinocytes and peripheral DRG sensory neurons (DRG SNs), however not with epidermal layer of differentiated and basal keratinocyte layers. Human skin equivalent (HSE) models have in vivo like epidermal layer under air-liquid interface culture condition, but lack the direct and sequential visualization of epidermal free nerve ending formation in 3D. F-11: F-11 cell line (a mouse N18TG2 neuroblastoma X rat DRG sensory neuron hybrid cell line), H: human, P: porcine, R: rat, M: mouse, Cap: capsaicin, 4α-PDD: 4α-phorbol 12,13-didecanoate, ATP: adenosine triphosphate, Mech: mechanical stimulation, Elec: electrical stimulation, Heat: Heat stimulation.

References

1. Le Pichon CE, Chesler AT. The functional and anatomical dissection of somatosensory subpopulations using mouse genetics. *Front Neuroanat* 8, 21 (2014).
2. Usoskin D, et al. Unbiased classification of sensory neuron types by large-scale single-cell RNA sequencing. *Nat Neurosci* 18, 145-153 (2015).
3. Smriti Iyengar, Michael H Ossipov, Kirk W Johnson, The role of calcitonin gene-related peptide in peripheral and central pain mechanisms including migraine. *Pain* 158, 543-559 (2017).
4. Koizumi S, Fujishita K, Inoue K, Inoue K, Shigemoto-Mogami Y, Shigemoto-Mogami Y, Tsuda M, Tsuda M, Inoue K, Inoue K. Ca²⁺ waves in keratinocytes are transmitted to sensory neurons: the involvement of extracellular ATP and P2Y₂ receptor activation. *Biochem J* 380, 329-338 (2004).
5. Leanne M Ramer, A Peter van Stolk, Jessica A Inskip, Matt S Ramer, Andrei V Krassioukov, Plasticity of TRPV1-Expressing Sensory Neurons Mediating Autonomic Dysreflexia Following Spinal Cord Injury. *Front Physiol* 3, 257 (2012).
6. Nozumi M, Togano T, Takahashi-Niki K, Lu J, Honda A, Taoka M, Shinkawa T, Koga H, Takeuchi K, Isobe T, Igarashi M, et al. Identification of functional marker proteins in the mammalian growth cone. *PNAS* 106, 17211-17216 (2009).
7. Sandilands A, Sutherland C, Irvine AD, McLean WH. Filaggrin in the frontline: role in skin barrier function and disease. *J Cell Sci* 122, 1285-1294 (2009).
8. Moreci RS, Lechler T. Epidermal structure and differentiation. *Curr Biol* 30, R144-R149 (2020).
9. Tominaga M, Takamori K. Itch and nerve fibers with special reference to atopic dermatitis: therapeutic implications. *J Dermatol* 41, 205-12 (2014)

Reviewers' Comments:

Reviewer #1:

Remarks to the Author:

I believe that the majority of the points raised regarding the development of the model and the additional experiments have been adequately addressed by the authors. The manuscript has significantly improved, however, there remain issues that must be addressed.

Title – I would suggest rephrasing the revised title as it is very confusing in its current form. For instance, it is not clear what “the innervated epidermis” refers to and what the model is.

Abstract – the authors refer to “sensory nerves” in the cultures. This is incorrect as nerves are tissues made up of axons, Schwann cells and many other cell types. This should say sensory axons.

Also “superior morphogenesis” should be rephrased as the comparison being made is unclear. Line 29- rephrase “platform for high-quality biomedical and pharmaceutical research.”

Introduction and Results

Although the introduction is now better balanced, it does primarily focus on the sensory neurons and the interaction with the epidermal layers. An important aspect of the study according to the authors is the anatomy of the recapitulated epidermal-like cell layers. The introduction should provide a brief overview of the structure of the epidermis.

Note that sensory neurons are pseudounipolar neurons and do not have dendrites. The reference to dendrites is confusing.

All references throughout the manuscript to “sensory nerves” must be replaced with the appropriate terminology. In culture, it would generally be sensory axons or neurites.

The references to “Adelta” and “C-fibers” in cultures should be avoided, or carefully explained. For instance, the authors do not provide evidence that the TRPV1+ axons that have crossed to the epidermal-like layers are of Adelta type. These designations as Adelta or C-fibers are relevant to the in vivo anatomy and best avoided when describing cell culture models.

In several places, the authors refer to the “protective effects” of the epidermal-like layer on the sensory neurons/axons. However, it is not at all clear how the layer has a protective effect on sensory neurons in culture. This should be clearly explained and backed up by evidence.

Figure 1a – “synaptic contact” in the epidermal layer is not relevant to the free nerve endings. This reference and the inset showing a “synaptic contact” should be removed.

Supplementary Figure 7 – The brightfield images are of very poor quality and should be replaced.

Reviewer #2:

Remarks to the Author:

Revision required

General

The manuscript has been extensively revised and improved

English needs editing. In the results section the quality of the epidermis is overstated and needs to be down-played a bit. Having said this, it is exceptional to be able to do so many readouts and to image whilst the culture is in the chip thus enabling cell cell interactions to be investigated without disrupting the the tissue by having to remove it the chip first. The neuronal-epidermal interactions are very impressive.

Introduction: supplementary table 1 is unclear as not all abbreviations and colours are explained. Importantly it is incomplete as many 3D skin models show e.g. loricrin (in stratum granulosum) and many authors have extensively studied barrier function in 3 D models.

The table also seems to indicate that the current manuscript is not so strongly novel as the authors would suggest.

Results

The quality of the epidermis is overstated and not better than 3D static models which are commercially available (MatTek, EpiSkin, CellSystems) or the multitude of inhouse models. Indeed Fig 3 images show extremely poor epidermal stratification and differentiation in the chip with no clear basal layer, stratum spinosum, granular layer or stratum corneum. The layers are disorganized as shown by no localized loricrin (stratum granulosum only), K10 (all suprabasal cells only) or K14 (basal layer only) expression as is repeatedly shown in 3D static models and skin.

This indicates that contrary to what the authors write, superior epidermis and barrier function is not achieved compared to many static 3D models. The claim about K14 basal, K10 suprabasal and loricrin granular expression is overstated. Of note, loricrin is never expressed in basal keratinocytes so it is not surprising that the planar-liquid model does not express it.

Thicker stratification is caused by air liquid exposure compared to wet / submerged exposure in 3D models. This then correlates to being less leaky and improved barrier. Of note, the condition of the planar/liquid cannot be used to represent current state of the art 3D static models as it is probably very wet (not air-exposed) due to leaking of medium through the hydrogel. This again questions how good this "novel" epidermal model really is.

Tilting of the chip only enabled the epidermis to remain dry as in standard 3D epidermal models, this in turn stimulates differentiation and improved barrier properties. This has been described by many others using the transwell system.

Page 6 line 148: you do not show that enhanced ERK activation increases proliferation only that ERK expression is increased. This claim is overstated.

The discussion is well written and the Methods section is clear to follow.

Response to Reviewer

General)

Before submitting our revised version of the manuscript, we would like to express our gratitude for the reviewers' insightful suggestions and insightful comments, which helped us to further improve our manuscript by elaborating our findings and statements.

Specific)

Reviewer-1

I believe that the majority of the points raised regarding the development of the model and the additional experiments have been adequately addressed by the authors.

The manuscript has significantly improved, however, there remain issues that must be addressed.

1. (Title) I would suggest rephrasing the revised title as it is very confusing in its current form. For instance, it is not clear what “the innervated epidermis” refers to and what the model is.

(Response)

Thank you for the recommendation. To improve comprehension, we revised the title;

(Revised title)

Reconstitution of innervated epidermis-mimicking layer on a microfluidic chip

2. (Abstract) the authors refer to “sensory nerves” in the cultures. This is incorrect as nerves are tissues made up of axons, Schwann cells and many other cell types. This should say sensory axons.

3. (Abstract) Also “superior morphogenesis” should be rephrased as the comparison being made is unclear.

4. (Abstract) Line 29- rephrase “platform for high-quality biomedical and pharmaceutical research.”

(Response)

We deeply appreciate the reviewer's comments. We revised the abstract as follows.

(Revised abstract)

Reconstruction of skin equivalents with physiologically relevant cellular and matrix architecture is indispensable for basic research and industrial applications. As skin-nerve crosstalk is increasingly recognized as a major element of skin physiological pathology, the development of reliable *in vitro* models is being

demanded. We developed a new microfluidic culture system to evaluate the selective communication between epidermal keratinocytes and sensory axons. Differentiated epidermal layer capable of barrier function was reconstituted in the slope-based air-liquid interfacing microfluidic chip. Spatial compartments enabled the formation of the physiologically relevant anatomy of the innervated epidermis and demonstrated the feasibility of *in situ* imaging and functional analysis in a cell-type-specific manner, thereby overcoming the limitations of conventional models. The developed system has the potential as an improved surrogate model and platform for biomedical research of neurocutaneous disorders or diabetic complications.

5. (Introduction and Results) Although the introduction is now better balanced, it does primarily focus on the sensory neurons and the interaction with the epidermal layers. An important aspect of the study according to the authors is the anatomy of the recapitulated epidermal-like cell layers. The introduction should provide a brief overview of the structure of the epidermis.

(Response)

We fully understand and agree with the reviewer's comments, and revised the manuscript as follows.

(Revised manuscript)

(Page 3, line 8) Free nerve endings of peptidergic or non-peptidergic C-fibers are mainly located close to keratinocytes in the spinous layer or granular layer of the epidermis, providing the structural basis for functional interaction such as synaptic-like contacts [9-12].

(Page 3, line 18) However, the traditional 2D coculture systems have failed to spatially locate a cell or cell portion (e.g., the axon and cell body of a neuron) and to selectively analyze and probe specific cells. Cultured keratinocytes also suffer from morphological and functional limitations [15, 17-18, 21]. The keratinocytes *in vivo* are existed in proliferating states at the basal layer of epidermis, and they undergo differentiation to form spinous, granular and cornified layer (Fig. 1a) [22]. 3D insert culture wells and microfluidic chips have been developed and further technologically improved by designing 3D culture conditions for epidermal morphogenesis and cell-customized compartmentalization for co-culture [13, 19-20, 23-26].

6. (Introduction and Results) Note that sensory neurons are pseudounipolar neurons and do not have dendrites. The reference to dendrites is confusing.

(Response)

We appreciated your helpful comment. We revised the manuscript as follows (Page 5, line 2).

(Revised manuscript)

Keratinocytes loaded into the epidermal channel grow on one side of extracellular matrix (ECM) hydrogel and interact only with axons but not with neuronal soma, enabling localized axon-keratinocyte interaction studies like *in vivo* physiology (Fig. 1a and c).

7. (Introduction and Results) All references throughout the manuscript to “sensory nerves” must be replaced with the appropriate terminology. In culture, it would generally be sensory axons or neurites.

(Response)

We deeply appreciate the reviewer's comments. We revised the manuscript as follows (Page 3, line 11 / Page 4, line 9 / Page 7, line 20 / Page 31, line 7).

(Revised manuscript)

(Page 3, line 11) Consistently with these physical contact, recent studies have shown that **sensory nerve fibers** in the skin can express and release nerve mediators ...

(Page 4, line 10) This work presents a new microfluidic model to coculture and analyzes 3D interactions of keratinocyte and **sensory neurons** *in vitro*.

(Page 7, line 21) PGP 9.5 + **sensory neurites** arise from the soma channel, penetrate the double-layered ECM hydrogels, move toward tortuous epidermis, ...

(Page 31, line 7) The number of sensory neurites (d) and the width of **sensory neurite bundles** ...

8. (Introduction and Results) The references to “Adelta” and “C-fibers” in cultures should be avoided, or carefully explained. For instance, the authors do not provide evidence that the TRPV1+ axons that have crossed to the epidermal-like layers are of Adelta type. These designations as Adelta or C-fibers are relevant to the in vivo anatomy and best avoided when describing cell culture models.

(Response)

We appreciated the reviewer's helpful comments. We revised the manuscript as follows (Page 9, line 13).

(Revised manuscript)

Increased number of **CGRP+ TRPV1+ fibers and TRPV1+ fibers** from co-cultured sensory neurons could also be the reason for the increased CGRP.

9. (Introduction and Results) In several places, the authors refer to the “protective effects” of the epidermal-like layer on the sensory neurons/axons. However, it is not at all clear how the layer has a protective effect on sensory neurons in culture. This should be clearly explained and backed up by evidence.

11. (Introduction and Results) Supplementary Figure 7 – The brightfield images are of very poor quality and should be replaced.

(Response)

We greatly value the reviewer's comments. We revised the supplementary figure 7 so that the morphology of the sensory neurons in the upper gel channel at 2h after 4 α -PDD treatment is more evident. 4 α -PDD was treated topically on the epidermal-like layer or directly to sensory neurons without epidermal-like layer. After 2h of treatment with 0.1 mM or 0.2 mM 4-PDD, sensory neurons without an epidermal-like layer exhibited morphological appearance of cell death, as depicted in supplementary figure 7a. However, sensory neurons with epidermal-like layer remained nearly intact, after treatment with 4-PDD.

We measured the molecular diffusion of fluorescein sodium (MW: 376.27 Da) through the epidermal-like layer (HEK layer). The size of fluorescein sodium was comparable to that of capsaicin (MW: 305.41 Da) and smaller than that of 4-PDD (MW: 672.9 Da). As depicted in supplementary figure 7b, the epidermal layers exhibited an apparent barrier function to fluorescein sodium.

(Revised supplementary figure 7)

10. (Introduction and Results) Figure 1a – “synaptic contact” in the epidermal layer is not relevant to the free nerve endings. This reference and the inset showing a “synaptic contact” should be removed.

(Response)

We fully understand and agree with the reviewer's comment. We revised the figure 1a as follows.

(Revised figure 1)

Reviewer: 2

The manuscript has been extensively revised and improved

English needs editing. In the results section the quality of the epidermis is overstated and needs to be down-played a bit. Having said this, it is exceptional to be able to do so many readouts and to image whilst the culture is in the chip thus enabling cell cell interactions to be investigated without disrupting the tissue by having to remove it the chip first. The neuronal-epidermal interactions are very impressive.

(Response)

Thank you for your feedback. The English editor carefully edited the revised manuscript. We've attached the editor's proof.

Carefully, an exaggeration regarding the quality of the epidermis was revised. The response will be added to the second question posed by the reviewer.

1. (Introduction) supplementary table 1 is unclear as not all abbreviations and colours are explained.

Importantly it is incomplete as many 3D skin models show e.g. loricrin (in stratum granulosum) and many authors have extensively studied barrier function in 3 D models.

The table also seems to indicate that the current manuscript is not so strongly novel as the authors would suggest.

(Response)

We greatly value the reviewer's insightful comments. Among the numerous *in vitro* skin models, we focused on the models that contained sensory neurons. As a result of the reviewer's suggestions, we conducted additional

research, added Table 1 to the supplementary materials, and revised the manuscript accordingly (Page 3, Introduction).

As the reviewer noted, barrier function has also been investigated in numerous 3D skin models including dermis mimicking layer with fibroblasts or endothelial cells, but not in 3D *in vitro* skin models containing sensory neurons. It is mainly due to the contraction of dermal layer, seriously damaging weak axons and neuron cells, and (if exist) their sensory receptors [Front Bioeng Biotechnol. 2020; 8: 388.]. This makes it difficult to analyze the sensory responses of the developed 3D innervated skin model via topical application. In our model, not only the epidermal differentiation and sensory nerve fiber specialization, but also the sensory neuronal response through an epidermis-resembling layer with barrier function, were evaluated successfully (Supplementary table 1).

(Page 3, Introduction)

However, the traditional 2D coculture systems have failed to spatially locate a cell or cell portion (e.g., the axon and cell body of a neuron) and to selectively analyze and probe specific cells. Cultured keratinocytes also suffer from morphological and functional limitations [15, 17-18, 21]. **The keratinocytes *in vivo* are existed in proliferating states at the basal layer of epidermis, and they undergo differentiation to form spinous, granular and cornified layer (Fig. 1a) [22].** 3D insert culture wells and microfluidic chips have been developed and further technologically improved by designing **3D culture conditions for epidermal morphogenesis and cell-customized compartmentalization for co-culture** [13, 19-20, 23-26]. In the 3D insert culture wells, a full-thickness human skin model with histological and functional properties that exhibit physiological similarity to *in vivo* skin was developed, but a reliable innervated skin model has yet to be reported [23-25, 27-31]. Recently reported sponge-based co-culture model, similar to the insert culture, also failed to mimic the anatomical distribution of intra-epidermal free nerve ending and axon patterning, **notwithstanding the well differentiated epidermal layer** (Supplementary Table 1) [20, 32-33].

(Revised supplementary table 1)

In vitro models of Sensory-Skin interaction			2D													this work
Reference number			[1]	[2]	[3]	[4]	[5]	[6]	[7]	[8]	[9]	[10]	[11]	[17]	2022	
Year			2004	2007	2009	2009	2010	2011	2012	2013	2012	2013	2014	2016	2022	
Cell type	Peripheral sensory neurons	DRG	M	R	R	M	P	R		P	P	R, M	R	P	R	
	Epidermal cells	Keratinocyte	H	R	R	M	P	H	H	P	H	R	H	H	H	
	Dermal cells	Fibroblast										H				
		Endothelial cell										H				
Other cells			A431, ND7-23	A431, ND7-23	HEK293			F-11			Atopic keratinocyte					
Phenotypic analysis	Pan-neuronal markers	PGP 9.5														
		Neurofilament														
		SMI 312														
		Peripherin														
	Peripheral sensory neuron subtypes	β-III tubulin (Tuj1)														
		NF200														
		TrkB														
		TRPV1														
		CGRP / SP														
	Histology	TrkA														
		IB4														
		Masson's trichrome														
		Hematoxylin & Eosin														
	Basal layer properties (epidermis)	Ki-67														
		K5														
		K14														
	Differentiated layer properties (epidermis)	K1														
		K10														
		Involucrin														
		Loricrin														
Receptor expression (epidermis)	Flaggrin															
	TRPV1															
Functional analysis	ELISA	CGRP concentration													Basal Cap 4α-PDD	
		SP concentration							Basal Cap	Basal Cap					Basal	
	Calcium imaging ATP imaging (A)	Mech Mech (A) ATP UTP	ATP Cap			ATP Heat (A)		Mech			Mech Cap		Cap		Cap	
Electrophysiology														Elec		
Permeability test (Barrier function)																

2D substrate
 Teflon divider
 Microfluidic device

In vitro models of Sensory-Skin interaction			3D													this work
Reference number			[6]	[12]	[13]	[14]	[15]	[16]	[17]	[18]	[19]	[20]	[21]	[22]	[23]	2022
Year			2003	2009	2009	2013	2014	2015	2016	2016	2017	2018	2019	2019	2022	2022
Cell type	Peripheral sensory neurons	DRG	R	M	M	P	M	M	P	P	R	H, M				R
	Epidermal cells	Keratinocyte	H	H	H	H	H	H	H	H	H	H	H	H	H	H
		Fibroblast		H	H	H	H	H	H	H	H	H	H	H	H	
		Endothelial cell		H	H		H	H								
Other cells				Schwann cell	Atopic keratinocyte						Diabetic fibroblast, keratinocyte		Schwann cell	hiNSC	hiNSC	hiNSC
	Pan-neuronal markers	PGP 9.5														
		Neurofilament														
SMI 312																
β-III tubulin (Tuj1)																
Peripheral sensory neuron subtypes	NF200															
	TRPV1															
	CGRP / SP															
	IB4															
	Histology	Masson's trichrome														
Hematoxylin & Eosin																
Basal layer properties (epidermis)	Ki-67															
	K5															
	K14															
Differentiated layer properties (epidermis)	K1															
	K10															
	Involucrin															
	Loricrin															
Receptor expression (epidermis)	Flaggrin															
	TRPV1															
Functional analysis	ELISA	CGRP concentration				Basal				Basal Cap		Basal Cap				Basal Cap 4α-PDD
		SP concentration				Basal	Basal					Basal Cap				Basal
	Calcium imaging	Mech							Menthol MPD		Cap					Cap
Permeability test (Barrier function)																

3D aggregate
 3D skin equivalent
 Microfluidic device

H; human, P; porcine, R; rat, M; mouse, hiNSC; human induced neural stem cell, Basal; unstimulated condition (basal)

level), Cap; capsaicin, 4 α -PDD; 4 α -phorbol 12,13-didecanoate, ATP; adenosine triphosphate, UTP; uridine triphosphate, Mech; mechanical stimulation, Elec; electrical stimulation, Heat; Heat stimulation. Colors; Yellow: 2D substrates culture model, Gray: Teflon divider culture model, Red: 3D aggregate model, Green: 3D skin equivalent model, Blue: Microfluidic chip culture model

2. (Results) The quality of the epidermis is overstated and not better than 3D static models which are commercially available (MatTek, EpiSkin, CellSystems) or the multitude of inhouse models. Indeed Fig 3 images show extremely poor epidermal stratification and differentiation in the chip with no clear basal layer, stratum spinosum, granular layer or stratum corneum. The layers are disorganized as shown by no localized loricrin (stratum granulosum only), K10 (all suprabasal cells only) or K14 (basal layer only) expression as is repeatedly shown in 3D static models and skin. This indicates that contrary to what the authors write, superior epidermis and barrier function is not achieved compared to many static 3D models. The claim about K14 basal, K10 suprabasal and loricrin granular expression is overstated. Of note, loricrin is never expressed in basal keratinocytes so it is not surprising that the planar-liquid model does not express it.

5. (Results) Page 6 line 148: you do not show that enhanced ERK activation increases proliferation only that ERK expression is increased. This claim is overstated.

(Response)

We value the reviewer's feedback. We concur with the reviewer that the developed skin layer structure in the microfluidic device is inferior to that of commercially available models. Please note, however, that the commercially available models have only epidermis. Dermis-integrated skin models for topically administered drugs are not yet available due to the severe contraction of the hydrogel-based dermis layer [Front Bioeng Biotechnol. 2020; 8: 388., Commun Biol. 2020; 3: 637.]. Our epidermis-like layer on the microfluidic chip is not superior, but it is innervated and suitable for testing drugs that are applied topically. Nonetheless, we removed the exaggerated claims you mentioned because it is evident that our epidermis-like layer is not yet perfect (Page 2 and 6-7).

(Revised abstract)

Reconstruction of skin equivalents with physiologically relevant cellular and matrix architecture is indispensable for basic research and industrial applications. As skin-nerve crosstalk is increasingly recognized as a major element of skin physiological pathology, the development of reliable *in vitro* models is being demanded. We developed a new microfluidic culture system to evaluate the selective communication between epidermal keratinocytes and sensory axons. Differentiated epidermal layer capable of barrier function was reconstituted in the slope-based air-liquid interfacing microfluidic chip. Spatial compartments enabled the formation of the physiologically relevant anatomy of the innervated epidermis and demonstrated the feasibility of *in situ* imaging and functional analysis in a cell-type-specific manner, thereby overcoming the limitations of conventional models. The developed system has the potential as an improved surrogate model and platform for biomedical research of neurocutaneous disorders or diabetic complications.

(Page 6-7, Results)

To model the sensory innervation to epidermis ^[11], we first adapted the slope-ALI method to induce the epidermal differentiation (Fig. 1b) ^[31, 38-39, 43]. Enhanced ERK activation of keratinocytes was observed at 30 min after slope-ALI culture, and 3 days of slope-ALI culture helped the keratinocytes to proliferate and finally to form thicker epidermal-like layers in histological (Fig. 3a-c, 3l-m, and Supplementary Fig. 5). Tilting angle of the chip after the exposure of keratinocytes to the ALI reduced the hydrostatic pressure to the keratinocyte layer to help the differentiated keratinocytes make better alignment (Fig. 1b and Supplementary Fig. 4). The epidermal-like layer expressed the markers of basal (cytokeratin 14⁺, K 14), suprabasal (cytokeratin 10⁺, K 10), and granular (loricrin⁺, for late-stage differentiation) layers. Keratinocytes cultured under planar-liquid condition mainly has basal keratinocytes without loricrin⁺ cells (Fig. 3d-h). ...

3. (Results) Thicker stratification is caused by air liquid exposure compared to wet / submerged exposure in 3D models. This then correlates to being less leaky and improved barrier. Of note, the condition of the planar/liquid cannot be used to represent current state of the art 3D static models as it is probably very wet (not air-exposed) due to leaking of medium through the hydrogel. This again questions how good this “novel” epidermal model really is.

4. (Results) Tilting of the chip only enabled the epidermis to remain dry as in standard 3D epidermal models, this in turn stimulates differentiation and improved barrier properties. This has been described by many others using the transwell system.

(Response)

The reviewer's remarks are certainly accurate. We utilized the air-liquid interface culture condition, which is already known to facilitate epidermal differentiation.

The first accomplishment of this research is the confirmation that a differentiated epidermal layer with barrier function can be reconstituted in a microfluidic channel. It permits real-time imaging, sample volume reduction, spatial patterning or compartmentalization of multiple cell types and hydrogels, and precise regulation of cell-cell communication and interaction in *in vitro* skin research. As the reviewer noted, we did not observe a fully differentiated epidermal layer in our model; however, we were able to evaluate the sensory innervation to the epidermal-like layer using a variety of histological and functional assays. As the second achievement, it may be sufficient to state that peptidergic sensory innervation was induced in the epidermal-like layer containing K10+ spinous layer, according to *in vivo* anatomical relationships [Neuron. 2005 Jan 6;45(1):17-25; Protein Cell. 2020 Apr;11(4):239-250].

For the tilting of the chip; the primary purpose of chip tilting is to maintain the epidermal channel's dryness. However, the tilling has an additional effect; the slight negative pressure exerted by the pressure head between the top of the medium in the soma channel and the epidermis channel, which was regulated by the chip's tilting angle, caused change in the morphology and internal structures of the epidermal-like layer. The increased contact between keratinocytes and ECM fibers as a result of negative pressure may be the cause of the change, but this

must be confirmed in future research. Due to the large volume of medium, such a minute negative pressure cannot be implemented in conventional *in vitro* skin models using transwell. We calculated the magnitude of the negative pressure and added supplementary figures 4e and f to illustrate its effect more clearly.

(Revised supplementary figure 4)

6. (Discussion and Methods) The discussion is well written and the Methods section is clear to follow.

(Response)

We appreciated the reviewer's comment. Thank you.

Reviewers' Comments:

Reviewer #1:

Remarks to the Author:

The manuscript by Ahn et al., describes the reconstitution of an epidermal layer, with the accompanying innervation by the sensory neurons, in a microfluidic cell culture platform. The authors further provide proof of concept for the versatility of this microfluidic platform for disease modelling and studying the interactions between the sensory neurons and keratinocytes. The manuscript makes a significant contribution to the tissue-on-the-chip approaches to in vitro disease modelling.

The authors have thoroughly addressed all my questions and comments and I recommend the publication of the manuscript.

Reviewer #2:

Remarks to the Author:

General

The manuscript is interesting but the novelty is still not clear compared to references in suppl table 1 and the epidermis quality is still extremely overstated.

Introduction:

Sup table 1 is much clearer but again this brings the issue of novelty. Ref 20 looks like it is just as good as the model proposed in this manuscript. What exactly is the difference between the models and what is superior about the model described in the manuscript? Also, what is superior about the model described in ref 20?

The authors answer in the rebuttal is not true "Please note, however, that the commercially available models have only epidermis." See e.g. MatTeks EpiDerm FT and Phenion FT skin models, in addition to many research grade models

"Technically, a slope-air liquid interface (slope-ALI) culture was applied to provide an air contact necessary for epidermal differentiation, demonstrating advancements in keratinocyte development in terms of epidermal morphogenesis, differentiation, and barrier function." This statement made by the authors is incorrect as indicated previously. This statement implies improvements in general, but this model has very inferior epidermal differentiation compared to static models in general. So what is it comparing to?

Results: Epidermal development in air-liquid interfacing microfluidic chip.

"The epidermal-like layer expressed the markers of basal (cytokeratin 14+, K 14), suprabasal (cytokeratin 10+, K 10), and granular (loricrin+, for late-stage differentiation) layers.

Keratinocytes cultured under planar-liquid condition mainly has basal keratinocytes without loricrin+ cells (Fig. 3d-h)." This text is not correct and is misleading. The K14, K10 and loricrin are intermitently expressed throughout the basal and suprabasal layers and not restricted to their correct skin locations. This indicates that stratification occurs but abnormal differentiation in their culture.

Text description of fig 4J is also incorrect. In the upper panel it appears that K14 (basal) may be above K10 (suprabasal which is opposite to lower image). The images do not show detailed enough histology to make any claims on epidermal quality or histology. This is not a superior epidermal culture as so emphasized throughout the manuscript.

Discussion:

"forming highly organized epidermal-like layer co-influencing with innervated sensory neurons." again is over stated regarding the epidermis

Response to Reviewer

Responses to Reviewer 1

The manuscript by Ahn et al., describes the reconstitution of an epidermal layer, with the accompanying innervation by the sensory neurons, in a microfluidic cell culture platform. The authors further provide proof of concept for the versatility of this microfluidic platform for disease modelling and studying the interactions between the sensory neurons and keratinocytes. The manuscript makes a significant contribution to the tissue-on-the-chip approaches to in vitro disease modelling.

The authors have thoroughly addressed all my questions and comments and I recommend the publication of the manuscript.

(Answer)

Thank you so much for your thoughtful comments. We deeply appreciate your valuable time for review procedure.

Responses to Reviewer 2

Q1) General: The manuscript is interesting but the novelty is still not clear compared to references in suppl table 1 and the epidermis quality is still extremely overstated.

(Answer)

We would like to begin by expressing our gratitude for reviewer 2's insightful suggestions and comments, which assisted us in refining our manuscript's findings and statements.

In this revision,

- We elaborated further on the three key points of reviewer 2's persistent questions regarding our paper: epidermal quality, novelty, and value of coculture system.
- We have also revised the description of the manuscript to make it more clear, including the title, abstract, introduction, results, and discussions, by actively considering the reviewer's comments (marked in blue in the manuscript). The revised parts will be displayed in response to question 2.

Q2) Introduction Q1: Sup table 1 is much clearer but again this brings the issue of novelty. Ref 20 looks like it is just as good as the model proposed in this manuscript. What exactly is the difference between the models and what is superior about the model described in the manuscript? Also, what is superior about the model described in ref 20?

(Answer)

In the previous revision, we clarified the distinction between our model and previous models by revising the abstract, introduction, and discussion to reflect our understanding and agreement with the reviewer's point. In this revision, we attempted to clarify the originality of our model in the abstract, introduction, and discussion.

The comparison between our model and the references on which the reviewer commented is provided below.

1. supplementary ref 20; [Acta Biomater. 82, 93-101 (2018)]

*** co-culture modeling:**

- The iPSC-derived sensory neurons (or mouse DRG neurons) and human keratinocytes are sequentially incorporated into the 3D collagen sponge, like insert culture system. And sponges were lifted up at air-liquid interface to promote keratinocyte differentiation for 17 days. ⇒

our system was differentiated for 5 days.

* Structural reconstitution:

- They assessed for sensory neurons by immunofluorescent staining of the markers BRN3A, β 3Tub, SP, NFM, TrkA and TRPA1 along with Ramp1 (a CGRP receptor) in sensory neuron-only cultured groups, but not the cocultured group. \Rightarrow composition and structural patterning of sensory neuronal subsets were not quantified. They failed to promote axonal outgrowth from the neuronal cell layer into the epidermal layer. There was no evidence of an anatomically innervated epidermal layer in their model.

Fig. 4. Characterization by immunofluorescence of the capacity of the iPSC-derived neurons to form a 3D nerve network in the tissue-engineered skin. The human reconstructed skin model made of keratinocytes (stained in green with Keratin 14), fibroblasts, and mouse (A) or iPSC-derived (B-D) neurons (stained in red with β 3Tub) was cocultured or not (B) with the mouse (C) or iPSC-derived Schwann cells (D). Scale bar = 50 μ m.

- They validated epidermal layer development with only K14 expression staining \Rightarrow cellular and structural development of the epidermal layer was not assessed. They have not verified the functional integrity (barrier function) of the epidermal layer. There was no evidence of the quality of the epidermal layer in their model.

* Functional integrity of sensory neurons and keratinocyte epidermal layer:

- They quantified SP and CGRP release from neurons in fibroblasts embedded collagen sponge, without epidermis by ELISA \Rightarrow The authors provided no obvious evidence for the transmission property of co-cultured neurons and keratinocytes. They did not directly verify the functional integrity and subtype-specific integration of the nerve cells and keratinocyte layer.

* Application of model:

- The authors described in the discussion ‘We intend to develop disease-specific skin models and analyze the potential effect of sensory nerves in these models after induction of neuropeptide release upon stimulation with TRPV1 agonists. Such a model would be particularly well-suited for the modeling of psoriasis, since this disease is highly suspected to be modulated by skin innervation. It represents an optimal base to build an immunocompetent skin model, with a strong potential for personalized medicine approaches.’, but provided no obvious evidence for their descriptions.

Fig. 2. Quantification of SP and CGRP release from iPSC-derived and mouse DRG neurons upon TRP stimulation. Substance P (SP; A-D) and calcitonin gene-related peptide (CGRP; E-H) released from iPSC-derived neurons (used after 19 days of maturation) (A,C,E,G) or mouse DRG neurons (B,D,F,H) were quantified by ELISA after stimulation with 65 nM potassium chloride (KCl, induces neuronal depolarization), 10 μ M capsaicin (CAP, ligand for TRPV1), 100 nM resiniferatoxin (RES, ligand for TRPV1), 1 mM eugenol (EUG, ligand for TRPV1,2,3) (except in D,H), 100 μ M allyl isothiocyanate (AITC, ligand for TRPA1), 100 ng/ml NGF (ligand for TrkA) cultured in monolayer (2D) or after seeding in a fibroblast-populated sponge (3D). Data are displayed as mean \pm standard error of the mean (* $p < 0.01$; ** $p < 0.001$; *** $p < 0.0001$, $n = 5$ to 21).

2. ref 20; [*Biomaterials* 113, 217-229 (2017)]

* co-culture modeling;

- They used a transwell insert to create a vertically stacked epidermal layer from keratinocytes on a hydrogel-based dermal layer containing fibroblasts.

Fig. 2. Setup for sensing measurements and structural characterization of HSE innervated with rat DRG neurons. Experimental setup design (A); immunofluorescence of Neurofilament-M (160 KDa) antibody expression (red signal) (white arrows indicated) and SHG signal in transversal cryo-section of innervated HSE (B, C, bars 25 mm); optical image of a bottom view of innervated HSE (D, bar 100 mm); immunofluorescence of Neurofilament-M (160 KDa) antibody expression in rat DRG neurons from the bottom tissue surface (E, bar 50 mm); DRG seeded in a Petri dish (F, bar 25 mm). DAPI staining was used for nuclei detection.

* Structural reconstitution;

- The authors displayed K10 and filaggrin expression in the monocultured keratinocyte layer and they validated epidermal layer development with two-photon reconstruction of the cross section of innervated HSE \Rightarrow cellular and structural development of the epidermal layer was not assessed. They have not verified the functional integrity (barrier function) of the epidermal layer. There was no evidence of the quality of the epidermal layer in their model.

- They assessed for sensory neurons by immunofluorescent staining of the marker NFM in the cocultured group. ⇒ composition and structural patterning of sensory neuronal subsets were not quantified. They failed to promote axonal outgrowth from the neuronal cell layer into the epidermal layer. There was no evidence of an anatomically innervated epidermal layer in their model.

Fig. 3. Calcium imaging study and HSE nerve sensory function. Immunohistochemical analysis performed on histological section of skin equivalent sample for HaCaT immunopositivity to TRPV1 antibody after 2 (A) and 8 (D) co-culture days with rat DRG neurons. Two-photon reconstruction of the cross section of innervated HSE showing neurofilament-M (red) and SHG signal of the collagen network (grey) at 2 (A) and 8 (B) days of co-culture. In B and E, the white arrows indicate the direction of the two-photon reconstruction; the dashed lines indicate the dermal/epidermal interfaces.

*** Functional integrity of sensory neurons and keratinocyte epidermal layer:**

- They analyzed calcium imaging on innervating HSE after the capsaicin addition without quantification of SP or CGRP release from neurons. ⇒ The authors provided no obvious evidence for the transmission property of co-cultured neurons and keratinocytes. They did not directly verify the functional integrity and subtype-specific integration of the nerve cells and keratinocyte layer.

Fig. 3. Calcium imaging study and HSE nerve sensory function. Graphs for calcium imaging study on innervated HSE model after 2 (C) and 8 (F) days of co-culture with rat DRG neurons before and after capsaicin addition on epidermal layer.

*** Application of model;**

- The authors provided no data.

⇒ In conclusion;

- Representative 3D innervated skin models in ref 20 and supplementary ref 20 were developed in conventional transwell insert like system, not in microfluidic chip. The authors did not sufficiently demonstrate the structural functional development of epidermal layer comparable to that of conventional transwell systems and the physical anatomy and signal transmission between epidermal keratinocytes and sensory neurons.

3. Our co-culture system

*** co-culture modeling;**

- We presented a 3D microfluidic model for coculture and analyzes 3D interactions of keratinocytes and sensory neurons *in vitro*. Technically, a slope-air liquid interface (slope-ALI) culture was applied to provide an air contact necessary for epidermal differentiation without additional insert devices, demonstrating advancements in keratinocyte development in terms of epidermal differentiation, cell layering, and barrier function compared to conventional microfluidic chip systems using planar liquid culture. It was also shown that the hydrogel-based multi-channel system recapitulated the cellular/subcellular compartmentalization and cell-cell/cell-matrix interactions, leading to the physiologically relevant organization of the innervated epidermal-like layer and enabling functional analysis in a cell-type-specific manner, such as the *in-situ* permeability assay of the epidermis and sensory transmission assay initiated by topical stimulation to epidermal keratinocytes.

* Structural reconstitution;

- Although not as perfect as the transwell insert system, our slope-ALI culture and coculture system generated a more stratified epidermal layer-like structure than the planar-liquid culture and ref 20, showing more intense, continuous distribution, and cellular layering K10+ suprabasal-like keratinocytes over the K14+ basal-like cells. We demonstrated the structural and functional quality of epidermal layers with various experimental methods.

- Although IB4⁺ neurons failed to recapitulate the complete anatomy of the native skin (not innervate into the deep epidermis), we recapitulated cellular and histological structures of the innervated epidermis more successfully than conventional 3D transwell insert culture or microfluidic culture system methods (Fig 3-5).

* Functional integrity of sensory neurons and keratinocyte epidermal layer;

- Cellular contacts and functional interaction between keratinocytes and neurons were observed in our innervated epidermal-like layers (Fig 5).

- We confirmed the nociceptive transduction by single-cell calcium imaging and SP/CGRP release quantification to evaluate the response of sensory neurons after topical treatment of (capsaicin, 4-PDD) on the epidermis (Fig. 3-5).

* Application of model;

- We modeled epidermal keratinocyte-sensory neuron crosstalk in our platform under hyperglycemic conditions to replicate hyperglycemia-induced diabetic neuropathy by simulating structural and functional changes in normal and disease states (Fig. 6) and demonstrated its feasibility as a model for investigating the underlying mechanism of the pathological condition.

⇒ In conclusion;

- Compared to the quality of the reconstituted epidermis layer, the 3D innervated skin models based on the conventional transwell system demonstrated insufficient sensory innervation data, which is considered a fundamental limitation of the transwell system from an analytical standpoint. We believed that the following features of the microfluidic system would enable it to overcome the aforementioned limitations: (transparency of materials, horizontally arranged channels, micro-scale culture chamber), by slope-ALI culture, optimized ECM hydrogels, and media in compartmented channels. We simulated 3D innervated epidermal

layer on a microfluidic chip, and investigated how keratinocytes forming epidermal-like layer and innervating sensory neurons influence each other, and innervated epidermal layer. After confirming the structural and functional integration of, disease modeling was feasible.

- Our system conceptually aimed to improve and recapitulate the innervated epidermal model structurally and functionally on the microfluidic chip, not the insert-based system. To overcome limitations of 2D or 3D coculture systems, our system enables histological recapitulation, such as subcellular compartmentalization of neurons and FNEs in epidermal layers, and functional analysis in a cell type-dependent manner or in situ functional assays such as Ca^{2+} transient and epidermal barrier testing. Moreover, we advanced the epidermal development on the chip by the slope-based culture to recapitulate the ALI culture, which was only possible in a 3D insert culture system.

Based on the comparison, we revised the manuscript as follows;

(revised manuscript ; marked blue in pages 3-4)

However, the traditional 2D coculture systems have failed to spatially locate a cell or cell portion (e.g., the axon and cell body of a neuron) and to selectively analyze and probe specific cells. Cultured keratinocytes also suffer from morphological and functional limitations [15, 17-18, 21]. The keratinocytes *in vivo* have existed in proliferating states at the basal layer of the epidermis, and they undergo differentiation to form a spinous, granular, and cornified layer (Fig. 1a) [22]. 3D transwell culture platforms and microfluidic chips have been developed and further technologically improved by designing 3D culture conditions for epidermal morphogenesis and cell-customized compartmentalization for co-culture [13, 19-20, 23-26]. In the 3D transwell insert culture system, a full-thickness human skin model with histological and functional properties that exhibit physiological similarity to *in vivo* skin was developed, but a reliable innervated skin model has yet to be reported [20, 23-25, 27-31]. A recently reported sponge-based co-culture model, like the transwell insert culture, also failed to mimic the anatomical distribution of intra-epidermal free nerve ending and axon patterning, notwithstanding the well-differentiated epidermal layer (Supplementary Table 1) [20]. The advantages of microfluidic chips, commonly referred to as lab-on-a-chip (LoC) or cell chips [19, 34], have made them attractive candidates to replace traditional experiments, by reducing the sample volume and the cost of reagents, and providing investigators with substantially precise control and predictability of the spatiotemporal dynamics of the cell microenvironments and fluids [19, 34]. In particular, the advantages of the spatiotemporal control allow researchers to closely recapitulate *in vivo* functions (both normal and disease states) by integrating several well-understood components into a single *in vitro* chip. However, reliable skin-nerve interactions and communication in the anatomically innervated epidermis have not yet taken advantage of microfluidics because they are based on the structure of vertically stacked systems, such as transwell insert cultures [16, 19, 35-36].

This work presents a new microfluidic model for coculture and analyzes 3D interactions of keratinocytes and sensory neurons *in vitro*. Technically, a slope-air liquid interface (slope-ALI) culture was applied to provide an air contact necessary for epidermal differentiation without additional devices, demonstrating advancements in keratinocyte development in terms of epidermal differentiation, cell layering, and barrier function compared to conventional microfluidic chip systems using planar liquid culture. It was also shown that the hydrogel-based multi-channel system recapitulated the cellular/subcellular compartmentalization and cell-cell/cell-matrix interactions, leading to the physiologically relevant organization of the innervated epidermal-like layer and enabling functional analysis in a cell-type-specific manner, such as the *in-situ* permeability assay of the epidermis and sensory transmission assay initiated by topical stimulation to epidermal keratinocytes. Finally, we modeled epidermal keratinocyte-sensory neuron crosstalk in our platform under hyperglycemic conditions mimicking acute diabetes and demonstrated its feasibility as a model for investigating the underlying mechanisms of the pathological condition.

(revised manuscript ; marked blue in pages 5)

This cellular compartmentalization allows two independent cells to be conducted on a single device maintaining cellular identity and function, and also allows to selectively analyze and/or probe specific cells and cell portions (e.g., the axon and cell body in a neuron) that cannot be done in 2D and transwell insert co-culture system (Fig. 1c). Each axon-guiding microchannel is individually filled by physiologically relevant ECM hydrogel, i.e., type 1 collagen, acting as a layer of acellular dermal ECM, yet exclusively without fibroblasts [22, 33, 37]. After seeding DRG neuron cells (in the soma channel) and human epidermal keratinocytes (HEK, in the epidermal channel) sequentially, the medium in the keratinocyte channels was emptied and the cell-filled chip was tilted to maintain above 30 degrees tilt to mimic the air-liquid interface (slope-ALI culture), a common and critical microenvironment for the skin cell differentiation (Fig. 1b, 1d, and Supplementary Fig. 1 and 4a-c) [31, 38-39]. The developed microfluidic chip enables various imaging, biochemical and functional analyses such as axonal response testing and integrity/permeability tests, which can be conducted directly on the innervated epidermis-on-chips, thus improving the limitations of conventional transwell insert culture or previous microfluidic culture systems (Fig. 1c).

(revised manuscript ; marked blue in pages 6-7)

To model the sensory innervation to the epidermis^[11], we first adapted the slope-ALI method to induce epidermal differentiation (Fig. 1b) [31, 38-39, 43]. Our slope-ALI method rapidly initiates ERK activation and the proliferation of keratinocytes than the planar-liquid method, resulting in thicker epidermal-like layers (Fig. 3a-c, 3k-l, and Supplementary Fig. 6). This method developed multicellular epidermal differentiation such as the basal (cytokeratin 14⁺, K 14), suprabasal (cytokeratin 10⁺, K 10), and granular (loricrin⁺, for late-stage differentiation) cells compared to the planar-liquid method: which consists mainly of K14⁺ keratinocytes but few K10⁺ and loricrin⁺ cells (Fig. 3d-f). The K14⁺ and K10⁺ keratinocytes of the slope-ALI method formed the suprabasal layer just above the basal layer like human epidermal tissue, showing a structurally more organized cell layer than the planar-liquid method (Fig. 3g). Under slope-ALI conditions, undulating micropatterned structures were noticed in the keratinocytes layer, like Rete ridge (RR) in natural human skin which has never been noticed in current tissue-engineered or 3D skin equivalents (Supplementary Fig. 7) [42]. The keratinocyte layer in slope-ALI condition was tortuous but tightly interconnected showing a strong barrier function to 3.984 kDa FITC-conjugated dextran, consistent with more intense and continuous distribution results (Fig. 3g). It also showed enhanced blocking for the diffusive transport from the epidermal channel to the soma channel, consequently facilitating cell-type-specific functional analysis (Fig. 1c). Taken together, these results indicate that our slope-ALI culture can accelerate the proliferation of keratinocytes and their aligned layering during differentiation, reconstituting the tortuous layered epidermal keratinocyte layer.

(revised manuscript ; marked blue in pages 8)

In the experiments, we found that NF200⁺ A-fibers (myelinated A-fibers) were more predominant than CGRP (peptidergic unmyelinated C-fibers) or IB4 (non-peptidergic unmyelinated C-fibers) positive neurons (Fig. 4f and 4h-i). NF200⁺ A-fibers from co-cultured SNs have morphologically thinner and longer than those from mono-cultured SNs and usually terminate in dermal ECMs falling short of the epidermal layer (Fig. 4f and 4h-i). CGRP⁺ peptidergic neurons were significantly more in co-cultured SNs and were mainly confined in the region under the epidermal layer, some terminated within the epidermis as free nerve endings. Whereas IB4⁺ non-peptidergic fibers from mono-cultured SNs had more quantity in ECM hydrogel. Although IB4⁺ neurons from co-cultured SNs migrated through ECM hydrogel relatively longer than those from mono-cultured SNs, they did not innervate into the deep epidermis (granular layer) and failed to recapitulate the complete anatomy of the native skin (Fig. 4f and 4h-i) [8-10]. Co-culture of SNs influence the development of epidermal keratinocytes in terms of morphogenesis and differentiation (Fig. 4j-k). When innervated, the epidermal-like layer grew on, not invading into the hydrogel, and presented enhanced alignment of K14, K10 and Loricrin (Supplementary Fig. 7a, 8a and 9a). In addition, the co-cultured epidermal keratinocyte layer showed a slight improvement in barrier function against 376.27 Da FITC-sodium (Fig. 4l-m). The co-culture of keratinocytes and sensory neurons in our slope-ALI microfluidic chip was proved to recapitulate cellular and histological structures of the innervated epidermis more successfully than

conventional 3D transwell insert culture or microfluidic culture system.

Figure 5 (page 8-9 and 32-33)

Figure 6 (page 9-10 and 34-35)

(Supplementary Table 1)

Supplementary Table 1. 2D or 3D in vitro models for co-culture of sensory neurons and keratinocytes.

In vitro models of Sensory-Skin interaction				2D													this work
Reference number				[1]	[2]	[3]	[4]	[5]	[6]	[7]	[8]	[9]	[10]	[11]	[17]		
Year				2004	2007	2009	2009	2010	2011	2012	2013	2012	2013	2014	2016		
Cell type	Peripheral sensory neurons	DRG	M	R	R	M	P	R			P	P	R, M	R	P	R	
	Dermal cells	Keratinocyte	H	R	R	M	P	H	H		P	H	R	H	H	H	
		Fibroblast										H					
		Endothelial cell										H					
Other cells				A431, ND7-23	A431, ND7-23	HEK293				F-11		Atopic keratinocyte					
Phenotypic analysis	Pan-neuronal markers	PGP 9.5															
		Neurofilament															
		SMI 312															
		Peripherin															
	Peripheral sensory neuron subtypes	β-III tubulin (Tuj1)															
		NF200															
		TrkB															
		TRPV1															
		CGRP / SP															
	Histology	Masson's trichrome															
		Hematoxylin & Eosin															
	Basal layer properties (epidermis)	Ki-67															
		K5															
		K14															
	Differentiated layer properties (epidermis)	K1															
		K10															
		Involucrin															
		Loricrin															
	Receptor expression (epidermis)	Flaggrin															
TRPV1																	
Functional analysis	ELISA	CGRP concentration										Basal Cap					
		SP concentration									Basal Cap						
	Calcium imaging ATP imaging (A)	Mech Mech (A) ATP UTP	ATP Cap				ATP Heat (A)			Mech			Mech Cap		Cap		
Electrophysiology																	
Permeability test (Barrier function)																	

2D substrate Teflon divider Microfluidic device

In vitro models of Sensory-Skin interaction				3D														this work
Reference number				[6]	[12]	[13]	[14]	[15]	[16]	[17]	[18]	[19]	[20]	[21]	[22]	[23]		
Year				2003	2009	2013	2014	2015	2016	2016	2017	2018	2019	2019	2022			
Cell type	Peripheral sensory neurons	DRG	R	M	M	P	M	M	P	P	R	H, M						
	Dermal cells	Keratinocyte	H	H	H	H	H	H	H	H	H	H	H	H	H			
		Fibroblast																
		Endothelial cell																
Other cells				Schwann cell	Atopic keratinocyte					Diabetic fibroblast, keratinocyte	Schwann cell	hNSC	hNSC	hNSC				
Phenotypic analysis	Pan-neuronal markers	PGP 9.5																
		Neurofilament																
		SMI 312																
		β-III tubulin (Tuj1)																
	Peripheral sensory neuron subtypes	NF200																
		TRPV1																
		CGRP / SP																
		IB4																
	Histology	Masson's trichrome																
		Hematoxylin & Eosin																
	Basal layer properties (epidermis)	Ki-67																
		K5																
		K14																
	Differentiated layer properties (epidermis)	K1																
		K10																
		Involucrin																
		Loricrin																
	Receptor expression (epidermis)	Flaggrin																
		TRPV1																
Functional analysis	ELISA	CGRP concentration				Basal			Basal Cap			Basal Cap						
		SP concentration				Basal	Basal					Basal Cap						
	Calcium imaging	Mech							Menthol MPD			Cap						
Permeability test (Barrier function)																		

3D aggregate 3D skin equivalent Microfluidic device

H; human, P; porcine, R; rat, M; mouse, hNSC; human induced neural stem cell, Basal; unstimulated condition (basal level), Cap; capsaicin, 4α-PDD; 4α-phorbol 12,13-didecanoate, ATP; adenosine triphosphate, UTP; uridine triphosphate, Mech;

mechanical stimulation, Elec; electrical stimulation, Heat; Heat stimulation. Colors; Yellow: 2D substrates culture model, Gray: Teflon divider culture model, Red: 3D aggregate model, Green: 3D skin equivalent model, Blue: Microfluidic chip culture model

Q3) Introduction Q2: The authors answer in the rebuttal is not true “Please note, however, that the commercially available models have only epidermis.” See e.g. MatTeks EpiDerm FT and Phenion FT skin models, in addition to many research grade models.

(Answer)

We apologize for the confusion caused by our prior response. We confirmed that dermis-integrated skin models are commercially available as the reviewer’s statement.

Q4) Introduction Q3: “ Technically, a slope-air liquid interface (slope-ALI) culture was applied to provide an air contact necessary for epidermal differentiation, demonstrating advancements in keratinocyte development in terms of epidermal morphogenesis, differentiation, and barrier function”. This statement made by the authors is incorrect as indicated previously. This statement implies improvements in general, but this model has very inferior epidermal differentiation compared to static models in general. So what is it comparing to?

(Answer)

Although our epidermal-like layer was incomparable to the epidermal layer of the transwell insert system, our slope-ALI culture and innervation model produced a more organized multicellular layer with enhanced barrier function than previous microfluidic models. Development of the epidermal layer was considerably accelerated compared to earlier microfluidic models, comparable to that on the transwell insert. In addition, unlike typical transwell systems, our innervated epidermal keratinocyte layer replicates the physical architecture between epidermal keratinocytes and sensory neurons (as described in previous responses).

- As described in the previous revised manuscript, we intended to directly compare our new model with each counterpart: slope-ALI cultured cells were compared with planar-liquid culture to demonstrate the effect of slope-ALI culture on epidermal keratinocyte differentiation, and sensory neurons-keratinocyte cocultured models were compared with keratinocyte monoculture to demonstrate the effect of innervation on epidermal keratinocyte differentiation in same slope-ALI culture condition. The most important aspect of our work is the qualitative enhancement of the SN-keratinocyte co-culture model relative to prior microfluidic chip or transwell co-cultures in terms of application, reliability, and

complexity that is not limited to epidermal layer differentiation. As stated by the reviewer, the quality of our epidermal layer is inferior to that of the transwell model, which has been acknowledged in the text as a limitation.

- However, we additionally conducted experiments to demonstrate that our epidermal-like layer was not too inferior to that of the transwell insert. The revision is summarized in the Q5 response.

Q5) Results: Epidermal development in air-liquid interfacing microfluidic chip.

Results Q1: “The epidermal-like layer expressed the markers of basal (cytokeratin 14+, K 14), suprabasal (cytokeratin 10+, K 10), and granular (loricrin+, for late-stage differentiation) layers. Keratinocytes cultured under planar-liquid condition mainly has basal keratinocytes without loricrin+ cells (Fig. 3d-h).” This text is not correct and is misleading. The K14, K10 and loricrin are intermittently expressed throughout the basal and suprabasal layers and not restricted to their correct skin locations. This indicates that stratification occurs but abnormal differentiation in their culture.

Results Q2: Text description of fig 4J is also incorrect. In the upper panel it appears that K14 (basal) may be above K10 (suprabasal which is opposite to lower image. The images do not show detailed enough histology to make any claims on epidermal quality or histology. This is not a superior epidermal culture as so emphasized throughout the manuscript.

(Answer)

About quality of epidermal layer, in general, epidermal keratinocytes progress from the basal to the cornified layers. KRT14 are typical for basal keratinocytes, while KRT10 are increasingly expressed during differentiation and postmitotic stages. The switch from KRT14 to KRT10 occurs in a gradual transition process. Therefore, a mixed composition of keratins can be observed in intermediate differentiation stages. In microfluidic chip using planar-liquid culture, K14+, K10+, or K14+K10+ (intermediate cells) epidermal keratinocytes were generated, but randomly distributed and not normally stratified like in vivo skin. However, although not as perfect as the transwell insert system, our slope-ALI culture also generated stratified keratinocyte layer-like structure, showing more intense, continuous distribution, and cellular layering K10+ suprabasal-like keratinocytes over the K14+ basal-like cells. We additionally added the representative data and revised the description for a clearer understanding of the epidermal keratinocyte layer development (marked in blue, page 6-7 and 27-28, Fig 3d-g).

- Figure 3d : Top & side view images of K10, K14 and Loricrin in epidermal-like layers formed under Planar-liquid and Slope-ALI conditions were included.

- Figure 3e : Quantified information for K14, K10 and Loricrin in fluorescent images were added.

About text description of fig 4j, we agreed that we were too biased in selecting representative images of each group. We added experiments and replaced the representative images and quantification data, and additionally revised the description for a clearer understanding of the epidermal keratinocyte layer development and comparative counterpart (marked in blue, page 8 and 29-30, Fig 4j-k).

- Figure 4j & 4k : Basal-suprabasal layer was noticed both in HEK mono (under Slope-ALI condition) and HEK+SN (under Slope-ALI condition) cases. The suprabasal layer in HEK+SN case is thicker than that in HEK mono, which was confirmed in the quantified data in figure 4k. The epidermal-like layer was formed on hydrogel layer in HEK+SN case, as contrast to the invading into the hydrogel in HEK mono case.

(revised manuscript ; marked blue in pages 6-7)

To model the sensory innervation to the epidermis^[11], we first adapted the slope-ALI method to induce epidermal differentiation (Fig. 1b)^[31, 38-39, 43]. Our slope-ALI method rapidly initiates ERK activation and the proliferation of keratinocytes than the planar-liquid method, resulting in thicker epidermal-like layers (Fig. 3a-c, 3k-l, and Supplementary Fig. 6). This method developed multicellular epidermal differentiation such as the basal (cytokeratin 14⁺, K 14), suprabasal (cytokeratin 10⁺, K 10), and granular (loricrin⁺, for late-stage differentiation) cells compared to the planar-liquid method: which consists mainly of K14⁺ keratinocytes but few K10⁺ and loricrin⁺ cells (Fig. 3d-f). The K14⁺ and K10⁺ keratinocytes of the slope-ALI method formed the suprabasal layer just above the basal layer like human epidermal tissue, showing a structurally more organized cell layer than the planar-liquid method (Fig. 3g). Under slope-ALI conditions, undulating micropatterned structures were noticed in the keratinocytes layer, like Rete ridge (RR) in natural human skin which has never been noticed in current

tissue-engineered or 3D skin equivalents (Supplementary Fig. 7) ^[42]. The keratinocyte layer in slope-ALI condition was tortuous but tightly interconnected showing a strong barrier function to 3.984 kDa FITC-conjugated dextran, consistent with more intense and continuous distribution results (Fig. 3g). It also showed enhanced blocking for the diffusive transport from the epidermal channel to the soma channel, consequently facilitating cell-type-specific functional analysis (Fig. 1c). Taken together, these results indicate that our slope-ALI culture can accelerate the proliferation of keratinocytes and their aligned layering during differentiation, reconstituting the tortuous layered epidermal keratinocyte layer.

(revised manuscript ; marked blue in pages 8)

Co-culture of SNs influence the development of epidermal keratinocytes in terms of morphogenesis and differentiation (Fig. 4j-k). When innervated, the epidermal-like layer grew on, not invading into the hydrogel, and presented enhanced alignment of K14, K10 and Loricrin (Supplementary Fig. 7a, 8a and 9a). In addition, the co-cultured epidermal keratinocyte layer showed a slight improvement in barrier function against 376.27 Da FITC-sodium (Fig. 4l-m). The co-culture of keratinocytes and sensory neurons in our slope-ALI microfluidic chip was proved to recapitulate cellular and histological structures of the innervated epidermis more successfully than conventional 3D transwell insert culture or microfluidic culture system.

(revised manuscript ; marked blue in pages 11)

Understanding the complex communications and interactions among various cells and neighboring microenvironmental components in the skin is essential for R&D and industrial applications, but challenges have existed in reconstituting 3D structures of cutaneous innervation *in vitro* and developing analysis tools in a cell-type-specific manner. The *in vitro* model can be not only an alternative to animals but also an approximation for various human skin diseases and side effects of other diseases on the skin. This paper describes a microfluidic co-culture system to form 3D innervated epidermal-like layers and its qualitative improvements in applicability, reliability, and complexity compared to previous microfluidic co-cultures. Precisely regulated spatial features and co-culture parameters allow compartmental patterning of neurons and epidermal keratinocytes, forming an organized innervated epidermal keratinocyte layer, being clearly visualized in microfluidic format.

Q6) Discussion: “forming highly organized epidermal-like layer co-influencing with innervated sensory neurons.” again is over stated regarding the epidermis

(Answer)

We thought that the clue for the co-influencing was presented in figures;

- organized epidermal-like layer in HEK+SN case
- morphological change of sensory neurons and formation of sensory receptors near HEK

However to minimize our overstatement, we performed additional experiments, added enhanced data and revised the whole manuscript as above.

(revised manuscript ; marked blue in pages 11)

Understanding the complex communications and interactions among various cells and neighboring microenvironmental components in the skin is essential for R&D and industrial applications, but challenges have existed in reconstituting 3D structures of cutaneous innervation *in vitro* and developing analysis tools in a cell-type-specific manner. The *in vitro* model can be not only an alternative to animals but also an approximation for various human skin diseases and side effects of other diseases on the skin. This paper describes a microfluidic co-culture system to form 3D innervated epidermal-like layers and its qualitative improvements in applicability, reliability, and complexity compared to previous microfluidic co-cultures. Precisely regulated spatial features and co-culture parameters allow compartmental patterning of neurons and epidermal keratinocytes, forming an organized innervated epidermal keratinocyte layer, being clearly visualized in microfluidic format.

Reviewers' Comments:

Reviewer #2:

Remarks to the Author:

The authors have revised the manuscript adequately and I propose acceptance for publication.